# The multifaceted role of EXOC6A in ciliogenesis

**Te-Lin Lin[1,2], Chien-Ting Wu[2,3], Tang K Tang[1,2]\***

[1]Taiwan International Graduate Program in Molecular Medicine, National Yang Ming Chiao Tung University and Academia Sinica, Taipei, Taiwan; [2]Institute of Biomedical Sciences, Academia Sinica, Taipei, Taiwan; [3]Department of Microbiology, University of Texas Southwestern Medical Center, Dallas, United States

## eLife Assessment

This **important** study elucidates the role of the exocyst component EXOC6A at distinct stages of ciliogenesis, which advances our understanding of ciliary membrane remodeling and cilium formation. The authors provide **compelling** evidence through high quality light and electron microscopic imaging, and careful analysis of knockout cell lines, that EXOC6A interacts with myosin-Va and is dynamically recruited via dynein-, microtubule-, and actin-dependent mechanisms, to support proper formation of the ciliary membrane. The study will be of interest to cell biologists and other researchers interested in vesicular trafficking, organellar membrane dynamics, and ciliogenesis.

**\*For correspondence:**
tktang@ibms.sinica.edu.tw

**Abstract** Ciliogenesis is a highly ordered process that requires membrane trafficking, fusion, and maturation. In this study, we investigated EXOC6A, a component of the exocyst complex known for secretory vesicle trafficking and fusion, and found that it interacts with myosin-Va (Myo-Va) during ciliogenesis. EXOC6A colocalizes with Myo-Va at various stages of ciliogenesis, including preciliary vesicles, ciliary vesicles (CVs), and ciliary sheath membrane during ciliogenesis. We found that EXOC6A vesicles are actively recruited, integrated, and exit from the CVs and the ciliary sheath, implying that EXOC6A vesicles may facilitate continuous cilia membrane remodeling during ciliogenesis. Importantly, EXOC6A knockout impairs ciliogenesis, arresting most cells at the CV stage and preventing recruitment of NPHP and MKS module components to the transition zone. Furthermore, EXOC6A vesicles are transported to the mother centriole via a dynein-, microtubule-, and actin-dependent mechanism. Our results suggest that EXOC6A functions in both early and late stages of ciliogenesis, and is involved in orchestrating vesicle dynamics, cilia membrane remodeling, and formation.

## Introduction

Most mammalian cells build primary cilia, which act as chemosensors or mechanosensors, and as transducers that regulate key developmental signaling pathways. Primary cilia are composed of nine doublet microtubules (MTs) arranged in a ring and surrounded by a ciliary membrane that is continuous with the plasma membrane (*Dawe et al., 2007*; *Nigg and Raff, 2009*). Primary cilia protrude from the cell surface to perform a wide range of biological functions. Defects in ciliogenesis can lead to a number of genetic disorders known as ciliopathies, which are characterized by loss of vision, disturbed kidney function, obesity, and defects in brain development (*Sattar and Gleeson, 2011*; *Valente et al., 2014*).

Ciliogenesis is a highly ordered and complex process. Currently, two distinct pathways have been reported, with different cell types involved: intracellular and extracellular. Polarized epithelial cells use

the extracellular pathway, whereas fibroblasts and mesenchymal cells use the intracellular pathway (*Breslow and Holland, 2019*). In the intracellular pathway, myosin-Va (Myo-Va) facilitates the transportation of preciliary vesicles (PCVs) to the distal appendages (DAs) of the mother centrioles (*Wu et al., 2018*). These PCVs may be derived from the Golgi apparatus (*Schmidt et al., 2012*) and subsequently fuse with a ciliary vesicle (CV) through the action of membrane fusion regulators EHD1 and EHD3 (*Lu et al., 2015*).

This step is critical for the conversion of the mother centriole into a basal body, which is responsible for removing the centriole cap protein, CP110, and recruiting ciliary transition zone (TZ) proteins (*Lu et al., 2015*). As a result, an axoneme extends within the larger CV, which gradually stretches out to form the double membranes of the ciliary shaft and sheath that surround the extending axoneme. Finally, the ciliary sheath (the outer membrane) fuses with the plasma membrane and the axoneme, enveloped by the ciliary shaft membrane, extends from the cell membrane and makes contact with the extracellular environment (*Ghossoub et al., 2011*). A signaling cascade, including RAB11, RABIN8, and RAB8, has also been reported to regulate the ciliary membrane assembly (*Chiba et al., 2013*; *Feng et al., 2012*; *Knödler et al., 2010*; *Nachury et al., 2007*; *Westlake et al., 2011*; *Yoshimura et al., 2007*). However, the processes of formation of the ciliary sheath and ciliary shaft membrane during ciliogenesis remain unclear.

The exocyst is a multi-subunit protein complex that was first identified in yeast, which mediates the tethering of secretory vesicles to the plasma membrane (*Mei and Guo, 2018*). The complex is composed of eight subunits, named Sec (for secretion; Sec3, Sec5, Sec6, Sec8, Sec10, Sec15, Exo70, and Exo84) in yeast or Exoc (for exocyst-related; EXOC1 to 8) in humans (*TerBush et al., 1996*; *TerBush and Novick, 1995*). The exocyst complex is implicated in various cellular processes, including exocytosis, ciliogenesis, cytokinesis, autophagy, cell polarity, migration, tumorigenesis, and fusion of secretory vesicles (*Mei and Guo, 2018*).

Mutations in genes encoding components of the exocyst complex have been associated with various human genetic disorders. For example, a mutation in *EXOC8* has been identified in Joubert syndrome (*Dixon-Salazar et al., 2012*), while a mutation in *EXOC4* has been reported in Meckel-Gruber syndrome (*Shaheen et al., 2013*). Both of these disorders involve clinically and genetically heterogeneous ciliopathy. Additionally, mutations in *EXOC2* cause severe defects in human brain development (*Van Bergen et al., 2020*), while mutations in *EXOC7* and *EXOC8* result in a novel disorder of cerebral cortex development, characterized by brain atrophy, seizures, developmental delay, and microcephaly (*Coulter et al., 2020*). However, the essential roles of these genes in ciliogenesis and their correlation with human genetic disorders are currently unknown.

It has been reported that some components of the exocyst complex are involved in ciliogenesis. For example, Sec10 (EXOC5) was shown to be required for ciliogenesis and cyst formation in vitro, and Sec10 knockdown leads to reduced ciliogenesis (*Zuo et al., 2009*; *Zuo et al., 2019*). Furthermore, other subunits such as EXOC3 and EXOC4 have been detected at the cilium with unknown function (*Seixas et al., 2016*). Interestingly, depletion of Sec8, Exo70, or Sec10 does not impair ciliogenesis (*Rivera-Molina et al., 2021*), whereas knockdown of Sec15a (EXOC6A) significantly perturbs ciliogenesis (*Rivera-Molina et al., 2021*). These findings imply that different exocyst subunits may have distinct, non-redundant functions during ciliogenesis. However, the precise localization of EXOC6A within cilia and its specific role in regulating ciliary membrane formation or trafficking events remain largely unknown.

Previously, we demonstrated that the earliest event in ciliogenesis is the transportation of PCVs by Myo-Va to the DAs of the mother centriole, and we showed that Myo-Va is released from the ciliary membrane upon fusion of the ciliary sheath with the plasma membrane (*Wu et al., 2018*). However, the mechanism of the assembly of double ciliary membranes (ciliary sheath and shaft) during intracellular ciliogenesis was left unknown. In this study, we provide evidence that the exocyst component EXOC6A interacts with Myo-Va and plays an essential role in ciliary membrane assembly at different stages during ciliogenesis.

## Results

### Colocalization patterns of EXOC6A and Myo-Va at PCVs, CVs, and ciliary sheath during intracellular ciliogenesis

To investigate the localization of EXOC6A during ciliogenesis, we treated retinal pigment epithelial (RPE1)-based mCherry-tagged Myo-Va cargo binding domain (Myo-Va-GTD)-inducible cells with doxycycline (Dox) for 24 hr and then serum-starved them for 2 hr to induce ciliogenesis. In this Tet-On system, Dox treatment triggers the inducible expression of the mCherry-Myo-Va-GTD transgene. We used antibodies against endogenous EXOC6A and glutamylated tubulin (Glu-tub), a marker of centrioles and ciliary axoneme, for confocal immunofluorescence analysis. Our results showed that the EXOC6A signal was mainly detected as a dot-shaped structure resembling a CV at the distal end of the mother centriole (*Figure 1A*, left), while some EXOC6A signals appeared to be colocalized with mCherry-tagged Myo-Va-GTD at the ciliary membrane (*Figure 1A*, right). To determine the subcellular localization of EXOC6A, we performed immunofluorescence staining using antibodies against CEP164 (a DA marker, *Graser et al., 2007*) and CEP120 (a centriole marker; *Lin et al., 2013*). The PCV/CV/ciliary sheath can be visualized with mCherry-tagged Myo-Va-GTD (*Wu et al., 2018*). The cells were then analyzed using three-dimensional structured illumination microscopy (3D-SIM; *Figure 1B*) or ultrastructure expansion microscopy (U-ExM; *Figure 1C*). Our results showed that EXOC6A signals colocalized with mCherry-Myo-Va-GTD at CV (*Figure 1B and C*). To further define the position of the EXOC6A signal, we performed correlative light and electron microscopy (CLEM) in RPE1-based GFP-EXOC6A-inducible cells (*Figure 1D*). Our CLEM results showed that the GFP-EXOC6A signals appeared to localize at PCVs (or vesicles), CVs (*Figure 1Di–ii*), and the elongating ciliary membrane (*Figure 1Diii–iv*).

To precisely analyze the subcellular localization of EXOC6A and Myo-Va during the formation of the ciliary sheath and the shaft membrane, we used 3D-SIM to analyze the RPE1 cells that exogenously express inducible GFP-EXOC6A using known markers combined for the ciliary sheath (Myo-Va-GTD; *Wu et al., 2018*), the ciliary shaft (ARL13B, INPP5E; *Feng et al., 2012*; *Garcia-Gonzalo et al., 2015*; *Wu et al., 2018*), and the axoneme (Glu-tub). EXOC6A signals were commonly detected at the Myo-Va-GTD-labeled ciliary sheath (*Figure 1Ei and ii*), which surrounds the ciliary shaft membrane (labeled ARL13B and INPP5E, *Figure 1Ei and iii*) and the axoneme (Glu-tub; *Figure 1Eii and iii*). Since our 3D-SIM analysis could not distinguish the spatial distances between INPP5E and the Glu-tub-labeled axoneme due to the limits of its resolution power (100–130 nm), we thus conclude that EXOC6A localizes at the ciliary sheath membrane surrounding the ciliary shaft. Altogether, our data show that EXOC6A, like Myo-Va, localizes at the PCVs, the CVs, and the ciliary sheath membrane.

### Spatial and temporal localization of Myo-Va and EXOC6A during ciliogenesis

To investigate the temporal and spatial correlation between the ciliary localization of EXOC6A and Myo-Va during ciliogenesis, we used RPE1-based mCherry-Myo-Va-GTD-inducible cells that were serum-starved, released at different time points, and analyzed using 3D-SIM with antibodies against endogenous EXOC6A and Glu-tub (*Figure 2A*). We identified five different phenotypes based on the immunostaining patterns of Myo-Va and EXOC6A during cilium assembly (*Figure 2B*), which are quantified in *Figure 2C*. Before serum starvation, a number of EXOC6A-associated vesicles were detected close to the mother centriole (*Figure 2A*, 0 min, type 1). During the progression of serum starvation, EXOC6A-associated vesicles appeared to gradually accumulate near the mother centriole (*Figure 2A*, 15 min, type 2) in a process we called 'clustering-to-centriole', and subsequently, EXOC6A signals were mainly detected at the Myo-Va-labeled CVs (*Figure 2A*, 30 min to 1 hr, type 3) and the ciliary sheath membrane, surrounding the Glu-tub-labeled axoneme (*Figure 2*, 2 hr, type 4). In some cells, a tubule-like structure (EXOC6A+/Myo-Va+) linked to the ciliary sheath membrane was observed (*Figure 2A*, 4 hr, white arrow). Finally, both EXOC6A and Myo-Va were released from the ciliary sheath membrane when it fused to the plasma membrane, while the Glu-tub-labeled axoneme continued to protrude into the extracellular milieu (*Figure 2A*, 24 hr, type 5). Interestingly, we observed that only a portion, but not all, of the EXOC6A-labeled vesicles colocalized with Myo-Va-labeled PCVs (*Figure 2A*, 0 and 15 min). However, once the CVs formed, both EXOC6A and Myo-Va-GTD signals were detected at the CVs (*Figure 2A*, 30 min, 45 min, and 1 hr). We present a schematic model in *Figure 2B* that shows

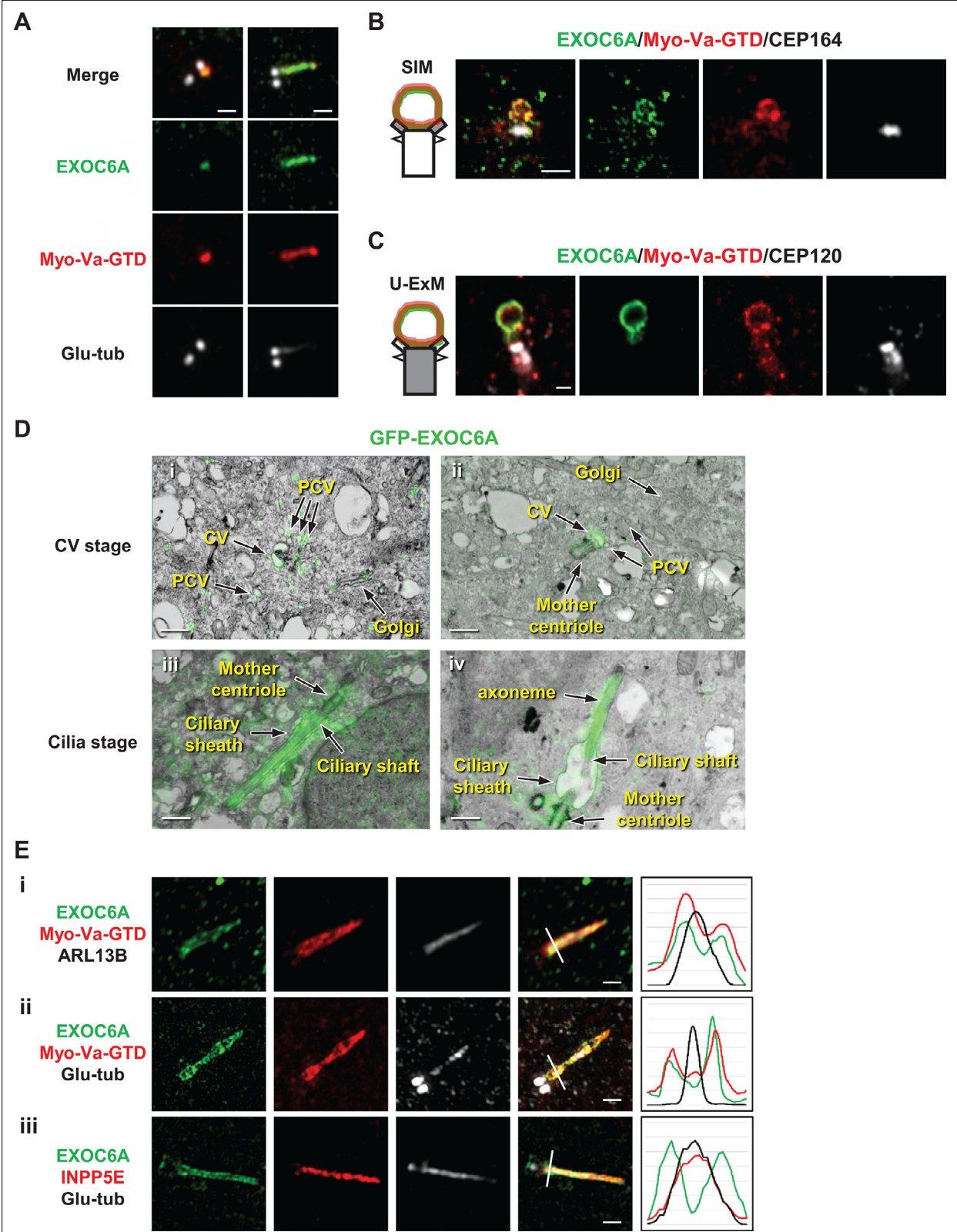

**Figure 1.** Colocalization patterns of EXOC6A and myosin-Va at preciliary vesicles (PCVs), ciliary vesicles (CVs), and ciliary sheath during ciliogenesis. (**A**) RPE1-based mCherry-Myo-Va-GTD cells were treated with doxycycline (Dox) for 24 hr, followed by serum starvation for 2 hr. Cells were then stained with antibodies against EXOC6A (green) and polyglutamylated tubulin (Glu-tub) (gray) and analyzed using immunofluorescence confocal microscopy. (**B, C**) RPE1-based mCherry-Myo-Va-GTD-inducible cells were treated as described in A and immunostained with antibodies against EXOC6A (green),

*Figure 1 continued on next page*

*Figure 1 continued*

CEP120 (centriole marker), and CEP164 (distal appendage marker, gray). Images were captured using three-dimensional structured illumination microscopy (3D-SIM) or ultra-expansion microscopy (U-ExM) with an LSM880 confocal system. (**D**) Correlative light and electron microscopy (CLEM) analysis of the localization of GFP-EXOC6A during ciliogenesis. RPE1-based GFP-EXOC6A-inducible cells were treated with Dox for 24 hr and subjected to serum starvation for 2 or 24 hr to observe the CVs or cilia membrane, respectively. Images were taken via SIM and TEM and then merged based on their relative localization. GFP-EXOC6A signals were located at PCVs, CVs (**i–ii**), and the ciliary membrane (**iii–iv**). (**E**) RPE1-based GFP-EXOC6A and mCherry-Myo-Va-GTD-inducible cells were treated with Dox for 24 hr and analyzed via 3D-SIM using the indicated antibodies. Right panels show fluorescence profile plots. EXOC6A colocalized with myosin-Va at the ciliary sheath, while ARL13B (**i**) and INPP5E (**iii**) are the ciliary shaft markers. Scale bars are 1 μm.

the localization of EXOC6A (green) and mCherry-Myo-Va-GTD (red) during ciliogenesis and the types of immunostaining patterns.

To further understand the relationship between the clustering-to-centriole progression of EXOC6A and Myo-Va during the early stage of ciliogenesis, we analyzed the intensity of both proteins clustering to the centrioles and calculated their overlapping coefficient correlation R (Pearson correlation coefficient) using ZEISS ZEN Blue software (*Figure 2—figure supplement 1*). We observed a gradual increase in the mean intensity of EXOC6A and Myo-Va clustering to the centrioles from 0 to 30 min (*Figure 2—figure supplement 1A–C*). Additionally, the correlation R between EXOC6A and Myo-Va significantly increased from 0 to 30 min (*Figure 2—figure supplement 1D*). These results suggest a high correlation between Myo-Va-labeled vesicles and EXOC6A-labeled vesicles during the progression of clustering-to-centriole.

## FRAP analysis of the dynamic localization of Myo-Va- and EXOC6A-labeled vesicles during ciliogenesis

We found that some EXOC6A structures had a tubule-like structure extending from the ciliary sheath (*Figure 2A*, 4 hr, type 4), suggesting the dynamic assembly of the ciliary membrane during ciliogenesis. To gain further insights into the dynamic relationship between Myo-Va- and EXOC6A-labeled vesicles during early CV formation and the later sheath membrane assembly stage, we performed live-cell imaging with fluorescence recovery after photobleaching (FRAP). FRAP is a widely used technique to measure the dynamics of ciliary membrane protein localization (*Kee et al., 2012*; *Westlake et al., 2011*). After photobleaching at CVs or the ciliary sheath region, the EXOC6A signals were recurrently recruited to the CVs (*Figure 3A*, *Figure 3—video 1*) and the ciliary sheath membrane within seconds (*Figure 3B*, *Figure 3—video 2*). Together, our data indicated that the EXOC6A-labeled vesicles are continuously recruited to and fused with the ciliary membrane not only at the CVs but also at the ciliary sheath. To further investigate the dynamic properties of the ciliary membrane, we performed live-cell super-resolution imaging using the ELYRA 7 SIM system. We used SiR-tubulin, a fluorescent probe derived from the MT-stabilizing drug docetaxel, to label endogenous MTs. This allowed us to visualize the structure of centrioles and axonemes during ciliogenesis in live imaging experiments (*Figure 3—figure supplement 1*). Our results show that EXOC6A colocalizes with Myo-Va-GTD on the ciliary membrane and surrounds the SiR-tubulin-labeled axoneme (*Figure 3—figure supplement 1A and B*). In addition, we detected a dynamic tubule-like structure that extended and retracted from the ciliary membrane (*Figure 3—figure supplement 1A and C* and *Figure 3—video 5*). To better observe vesicle trafficking and membrane dynamics, we used the single-channel and burst modes of the ELYRA 7 SIM system to achieve higher temporal resolution in live imaging. Our results showed that GFP-EXOC6A-labeled vesicles gradually moved to and fused with CVs (*Figure 3C*, white arrow, *Figure 3—video 3*) and the ciliary pocket, a membrane domain found at the base of primary cilia (*Figure 3D*, white arrow, *Figure 3—video 4*). Interestingly, almost at the same time, some GFP-EXOC6A-labeled vesicles were fused with or secreted from the ciliary pocket (*Figure 3D*, red arrow, *Figure 3—video 4*). Unexpectedly, we observed that some GFP-EXOC6A-associated tubular structures (*Figure 3D*, yellow arrow) or GFP-EXOC6A-labeled vesicles were released directly from the ciliary membrane rather than from the ciliary pocket (*Figure 3D*, orange arrow). Together, our results indicate that GFP-EXOC6A-associated vesicles/membranes are highly dynamic and continuously recruited to and/or released from CVs during the early stages of ciliogenesis, and that they are also fused with or excreted from the ciliary membrane during later stages of ciliogenesis. We, therefore, propose that EXOC6A vesicles are

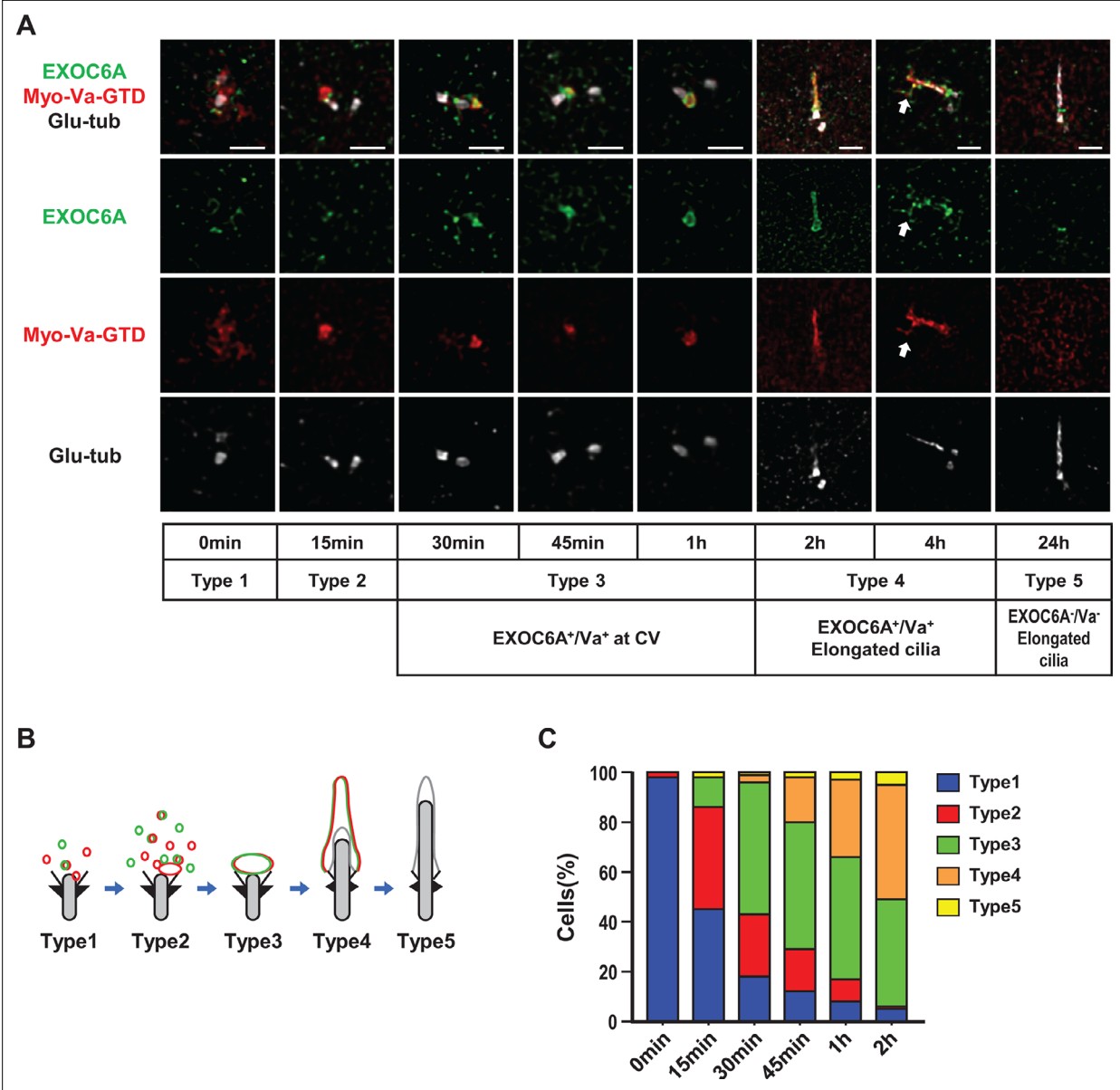

**Figure 2.** Spatial-temporal localization of EXOC6A and myosin-Va (Myo-Va) during ciliogenesis. (**A**) RPE1-based mCherry-Myo-Va-GTD-inducible cells were treated with doxycycline (Dox) for 24 hr, serum-starved, released at different time points, and analyzed via three-dimensional structured illumination microscopy (3D-SIM) using the indicated antibodies. The white arrow indicates a tubule extending from the ciliary membrane. (**B, C**) Schematic model showing the localization patterns of EXOC6A (green) and Myo-Va (red) during ciliogenesis, and the percentages of cells with different types of staining patterns (%) are shown in C. Scale bars are 1 μm.

The online version of this article includes the following figure supplement(s) for figure 2:

**Figure supplement 1.** Correlation of spatial localization of EXOC6A and myosin-Va (Myo-Va) during the early stages of ciliogenesis.

responsible for the exchange of materials within existing CVs and the ciliary sheath during ciliogenesis, facilitating the addition or removal of membranes or membrane proteins.

## Myo-Va and EHD1 are required for EXOC6A-labeled CV formation, and EXOC6A is associated with Myo-Va

We previously reported that Myo-Va-mediated transportation of PCVs to the mother centriole is the earliest event that defines the onset of ciliogenesis (*Wu et al., 2018*). In this study, we aimed to examine the sequential order of PCV transportation mediated by Myo-Va and EXOC6A during ciliogenesis in

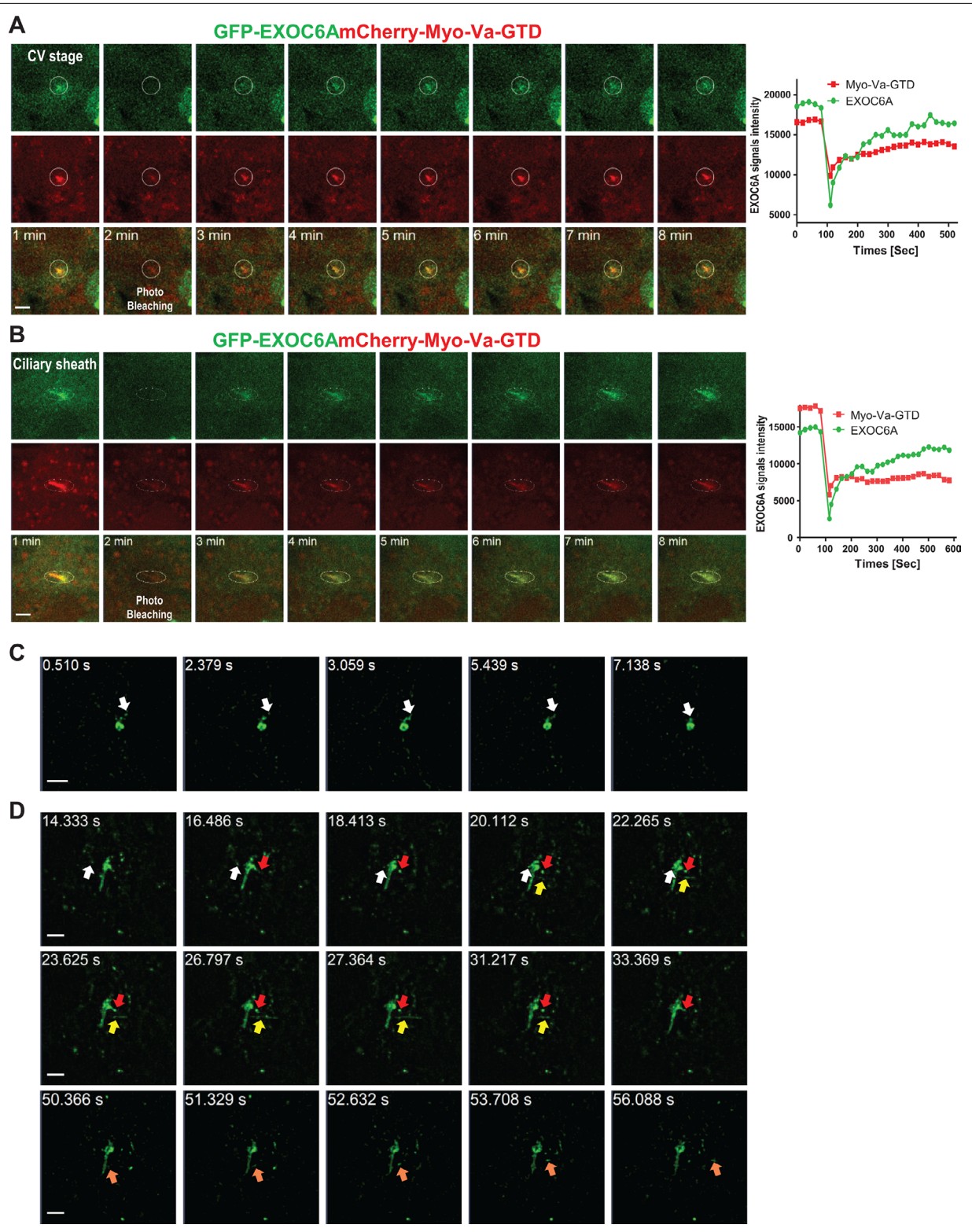

**Figure 3.** Fluorescence recovery after photobleaching (FRAP) analysis of Myo-Va- and EXOC6A-labeled vesicles during ciliogenesis. (**A, B**) RPE1-based GFP-EXOC6A and mCherry-Myo-Va-GTD-inducible cells were treated with doxycycline (Dox) for 24 hr and serum-starved for 30 min. Regions of ciliary vesicles (CVs) (**A**) or the ciliary membrane (**B**) of the cells were photobleached using the LSM880 confocal microscope with the 405 nm laser (related to *Figure 3—videos 1 and 2*). Signal intensity of EXOC6A is shown in the right panel. (**C, D**) Dynamic localization of GFP-EXOC6A at CVs (**C**) or the ciliary membrane (**D**) during ciliogenesis. RPE1-based GFP-EXOC6A-inducible cells were treated with Dox for 24 hr and serum-starved for 30 min. Images were

*Figure 3 continued on next page*

*Figure 3 continued*

taken using an Elyra 7 high-speed live-cell imaging system. Series of images in (**C**) shows the progression of fusion of GFP-EXOC6A-associated vesicles into CVs (white arrow). Series of images in (**D**) shows the process by which GFP-EXOC6A-associated vesicles fuse into the ciliary pocket (white arrow) and simultaneously exit or reintegrate into the ciliary pocket (red arrow). Some images also show the release of tubule-like structures in the ciliary membrane (yellow arrow) or the excretion of vesicles in the ciliary membrane (orange arrow). These data suggested that the components of CVs, the ciliary pocket, and the ciliary membrane are highly dynamic (related to *Figure 3—videos 3 and 4*). Scale bars are 2 µm.

The online version of this article includes the following video and figure supplement(s) for figure 3:

**Figure supplement 1.** Live-cell imaging of GFP-EXOC6A colocalized with Myo-Va-GTD at the ciliary membrane.

**Figure 3—video 1.** Myo-Va-GTD were treated with doxycycline (Dox) for 24 hr and then serum starved for 30 min.

https://elifesciences.org/articles/108271/figures#fig3video1

**Figure 3—video 2.** RPE1-based inducible cells expressing GFP-EXOC6A and mCherry-Myo-Va-GTD were treated with doxycycline (Dox) for 24 hr and then serum-starved for 30 min (related to *Figure 3B*).

https://elifesciences.org/articles/108271/figures#fig3video2

**Figure 3—video 3.** RPE1-based inducible cells expressing GFP-EXOC6A were treated with doxycycline (Dox) for 24 hr and then serum-starved for 30 min (related to *Figure 3C*).

https://elifesciences.org/articles/108271/figures#fig3video3

**Figure 3—video 4.** RPE1-based inducible cells expressing GFP-EXOC6A were treated with doxycycline (Dox) for 24 hr and then serum-starved for 30 min (related to *Figure 3D*).

https://elifesciences.org/articles/108271/figures#fig3video4

**Figure 3—video 5.** RPE1-based inducible cells expressing GFP-EXOC6A and mCherry-MyoVa-GTD were treated with doxycycline (Dox) for 24 hr, then serum-starved for 30 min, and SiR-tubulin was added to label centriole and axoneme (related to *Figure 3—figure supplement 1*).

https://elifesciences.org/articles/108271/figures#fig3video5

both RPE1-based *Myo-Va* knockout (KO) and *EXOC6A* KO cells. We constructed two independent *EXOC6A* KO cell lines using the CRISPR-Cas9 system. The success of the gene disruption was validated at the genomic level through target site sequencing (*Figure 4—figure supplement 1A*), a schematic diagram of predicted peptide products (*Figure 4—figure supplement 1B*), and western blot analysis (*Figure 4—figure supplement 1C*), confirming the absence of EXOC6A expression in all KO cell lines. All independent KO cell lines exhibited similar ciliary generation defects (*Figure 4—figure supplement 1D*). The EXOC6A signal is located at CV (*Figure 4—figure supplement 1E*) and cilia sheath (*Figure 4—figure supplement 1F*), but is absent in both *EXOC6A* KO cell lines. Our results showed that few EXOC6A-labeled vesicles and no EXOC6A-labeled CVs were detected at the distal end of the mother centriole in the *Myo-Va* KO cells (*Figure 4A*). In contrast, Myo-Va-labeled CVs were frequently observed at the mother centrioles of *EXOC6A* KO cells (*Figure 4B*). Similar results were also observed in the other *EXOC6A* KO cell line. Together, our findings suggest that EXOC6A is not essential for the formation of Myo-Va-labeled CVs during ciliogenesis.

EHD1 was reported to be essential for the fusion of PCVs to form larger CVs during the early stage of ciliogenesis (*Lu et al., 2015*). To investigate the sequential order of EHD1- and EXOC6A-mediated ciliary membrane assembly, we depleted EHD1 expression using siRNA in RPE1 cells. Our results showed that EHD1 depletion led to a reduction in the formation of EXOC6A-labeled CVs (*Figure 4C*). In contrast, *EXOC6A* KO did not appear to affect EHD1-mediated fusion of PCVs to form a large EHD1-labeled CV at the mother centriole (*Figure 4D*). Together, our findings suggest that EHD1 is required for EXOC6A-labeled CV formation, while EXOC6A is not essential for the formation of EHD1-labeled CVs.

Additionally, previous studies reported that Myo2p, the yeast homolog of human Myo-Va, interacts with sec15, the yeast homolog of human EXOC6A (*Jin et al., 2011*). The present study showed that some, but not all, of the EXOC6A-labeled PCVs colocalized with Myo-Va-labeled PCVs (*Figure 2A*, 0 and 15 min). We speculate that Myo-Va may transport some EXOC6A-labeled vesicles to the CVs for ciliary membrane assembly. Our co-immunoprecipitation experiments showed that endogenous EXOC6A can co-immunoprecipitate with Myo-Va, supporting this hypothesis (*Figure 4E*).

Finally, previous studies have indicated that the removal of centriolar protein CP110 from the distal end of the mother centriole is necessary for axoneme growth (*Spektor et al., 2007*). It has been proposed that CP110 removal occurs after the docking of CV to the DA of the mother centriole (*Lu et al., 2015*; *Wu et al., 2018*). To investigate the impact of EXOC6A loss on CP110 removal, we

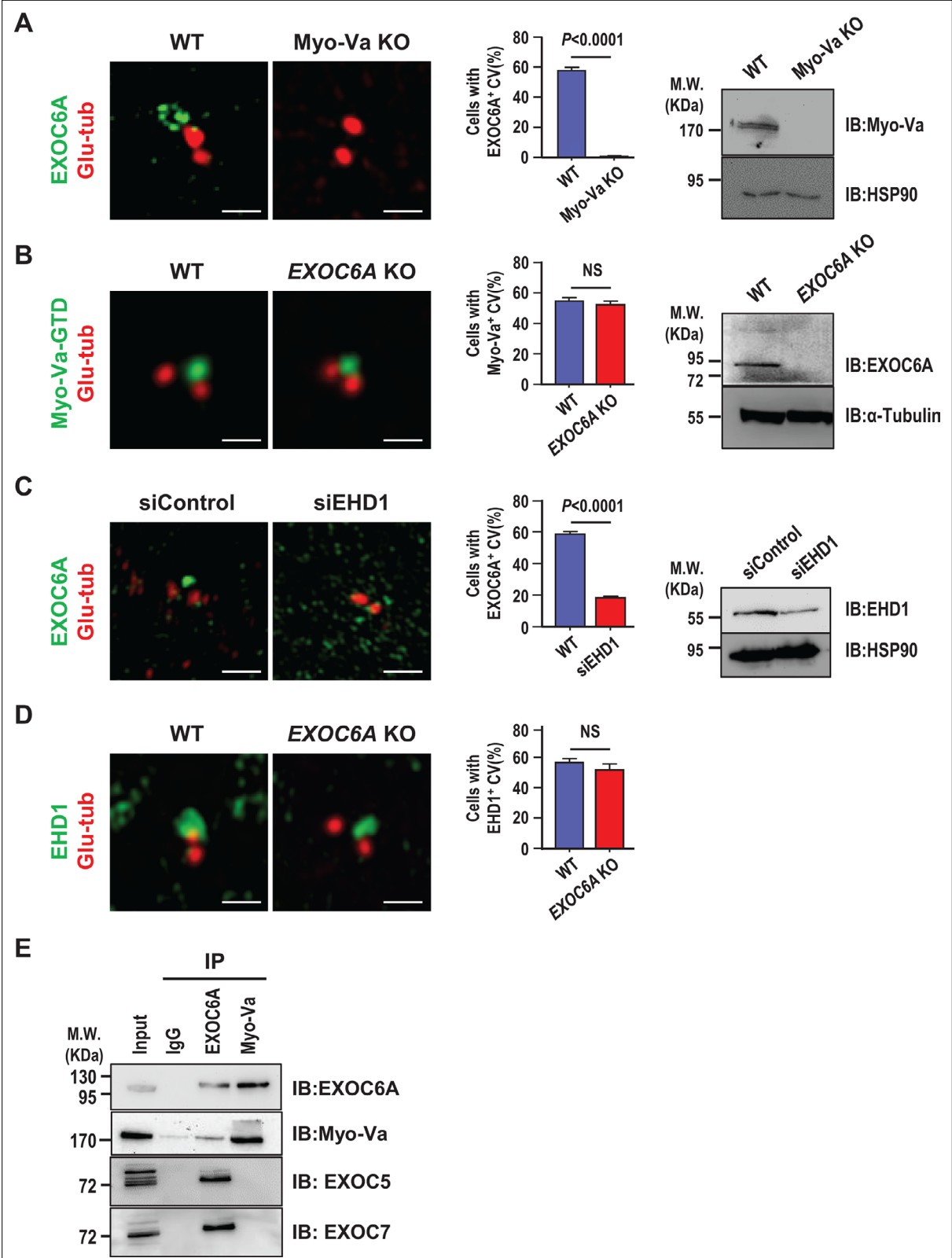

**Figure 4.** Myosin-Va (Myo-Va) and EHD1 are required for EXOC6A-labeled ciliary vesicle (CV) formation, and EXOC6A is associated with Myo-Va. (**A**) Myo-Va is required for EXOC6A-labeled CV formation. RPE1-based *Myo-Va* knockout (KO) cells and wild-type (WT) cells immunostained with antibodies against EXOC6A (green) and glutamylated tubulin (Glu-tub) (red). Quantification of cells with EXOC6A-positive CVs (middle panel) and immunoblotting results (right panel) are shown. (**B**) EXOC6A is not required for Myo-Va-labeled CV formation. RPE1-based mCherry-Myo-Va-GTD-

*Figure 4 continued on next page*

*Figure 4 continued*

inducible cells against *EXOC6A* KO background and WT cells were immunostained with antibodies against Glu-tub (red). For easy differentiation, the mCherry-Myo-Va-GTD signal is converted to green. Quantification of cells with Myo-Va-associated CVs (middle panel) and immunoblotting results (right panel) are shown. (C) EHD1 is required for EXOC6A-labeled CV formation. RPE1 cells treated with siControl or siEHD1 for 48 hr were immunostained with antibodies against EXOC6A (green) and Glu-tub (red). Quantification of cells with EXOC6A-positive CVs (middle panel) and immunoblotting results (right panel) are shown. (D) EXOC6A is not required for EHD1-labeled CV formation. RPE1-based *EXOC6A* KO and WT cells were immunostained with antibodies against EHD1 (green) and Glu-tub (red). Quantification of cells with EHD1-positive CVs is shown (right panel). (E) Co-IP experiments analyzed the association between EXOC6A and its potential binding proteins. Cell lysates were immunoprecipitated (IP) with EXOC6A or Myo-Va antibodies, followed by immunoblotting with the indicated antibodies. Error bars in A–D represent mean ± s.d. from at least 3 independent experiments with 100 cells per experiment. p-Value was determined with two-tailed Student's *t*-test. p<0.05 was considered statistically significant. NS, not significant. Scale bars are 1 μm.

The online version of this article includes the following source data and figure supplement(s) for figure 4:

**Source data 1.** PDF file containing original western blots for *Figure 4*, indicating the relevant bands and treatments.

**Source data 2.** Original files for western blot analysis displayed in *Figure 4*.

**Figure supplement 1.** Construction and characterization of EXOC6A knockout (KO) cell lines.

**Figure supplement 1—source data 1.** PDF file containing original western blots for *Figure 4—figure supplement 1C*, indicating the relevant bands and treatments.

**Figure supplement 1—source data 2.** Original files for western blot analysis displayed in *Figure 4—figure supplement 1C*.

**Figure supplement 2.** *EXOC6A* deletion does not interfere with the removal of CP110 from the mother centriole.

---

examined CP110 localization in *EXOC6A* KO cells. Our results showed that CP110 is appropriately removed from the distal end of the mother centriole in *EXOC6A* KO cells, indicating that EXOC6A is not required for this process (*Figure 4—figure supplement 2*), which further supports that EXOC6A is not essential for CV formation. Taken together, our findings suggest that Myo-Va transports a subset of EXOC6A-labeled vesicles to the existing CVs of the mother centriole for ciliary membrane assembly. However, EXOC6A is not essential for CV formation.

## Transportation of EXOC6A-labeled vesicles to the mother centriole is via a dynein-, MT-, and Arp2/3-dependent pathway

Our previous study showed that the transportation of PCVs to the DAs of the mother centriole is mediated by Myo-Va, dynein, MTs, and the Arp2/3 complex (*Wu et al., 2018*). In the current study, we found that Myo-Va and EHD1 are required for EXOC6A-labeled CV formation and that Myo-Va co-immunoprecipitates with EXOC6A (*Figure 4*). It has been reported that dynein is necessary for the correct positioning of the Golgi complex near the centrosome upon serum starvation (*Palmer et al., 2009*; *Wu et al., 2018*), and that the inhibition of dynein activity with ciliobrevin D suppresses ciliogenesis (*Firestone et al., 2012*; *Wu et al., 2018*). Therefore, we investigated whether the transport of EXOC6A-associated vesicles to the DA of the mother centriole also requires dynein, MTs, and the Arp2/3 complex.

RPE1 cells were transfected with siRNAs targeting Golgin160, a protein required for the recruitment of dynein to the Golgi complex (*Yadav et al., 2012*), or treated with ciliobrevin D, a known cytoplasmic motor dynein inhibitor that suppresses ciliogenesis (*Firestone et al., 2012*), following the protocol described in *Figure 5A*. Our findings showed that depletion of Golgin160 led to a decrease in EXOC6A-labeled CV signals at or near the mother centriole (*Figure 5B and D*) and resulted in the dispersion of Golgi matrix protein 130 (GM-130)-labeled Golgi complexes (*Figure 5B and E*), ultimately leading to a reduction in the total number of cilia (*Figure 5F*). Similar results were also observed in RPE1 cells treated with 10 μM ciliobrevin D (*Figure 5Ci and D–F*). We next treated RPE1 cells with 10 μM nocodazole (NZ, an MT-depolymerizing drug) and examined its effect on ciliogenesis. Our results showed that the EXOC6A-labeled CVs and the cilia number are greatly reduced in NZ-treated cells (*Figure 5Cii, D, and F*). These data suggest that a subset of EXOC6A-positive vesicles may originate from the Golgi apparatus. Furthermore, the accumulation of these vesicles at the mother centrioles is highly sensitive to disruption of dynein or MTs, suggesting that efficient transport of these vesicles may depend on the integrity of the MT network. However, more experiments are required to confirm this conclusion.

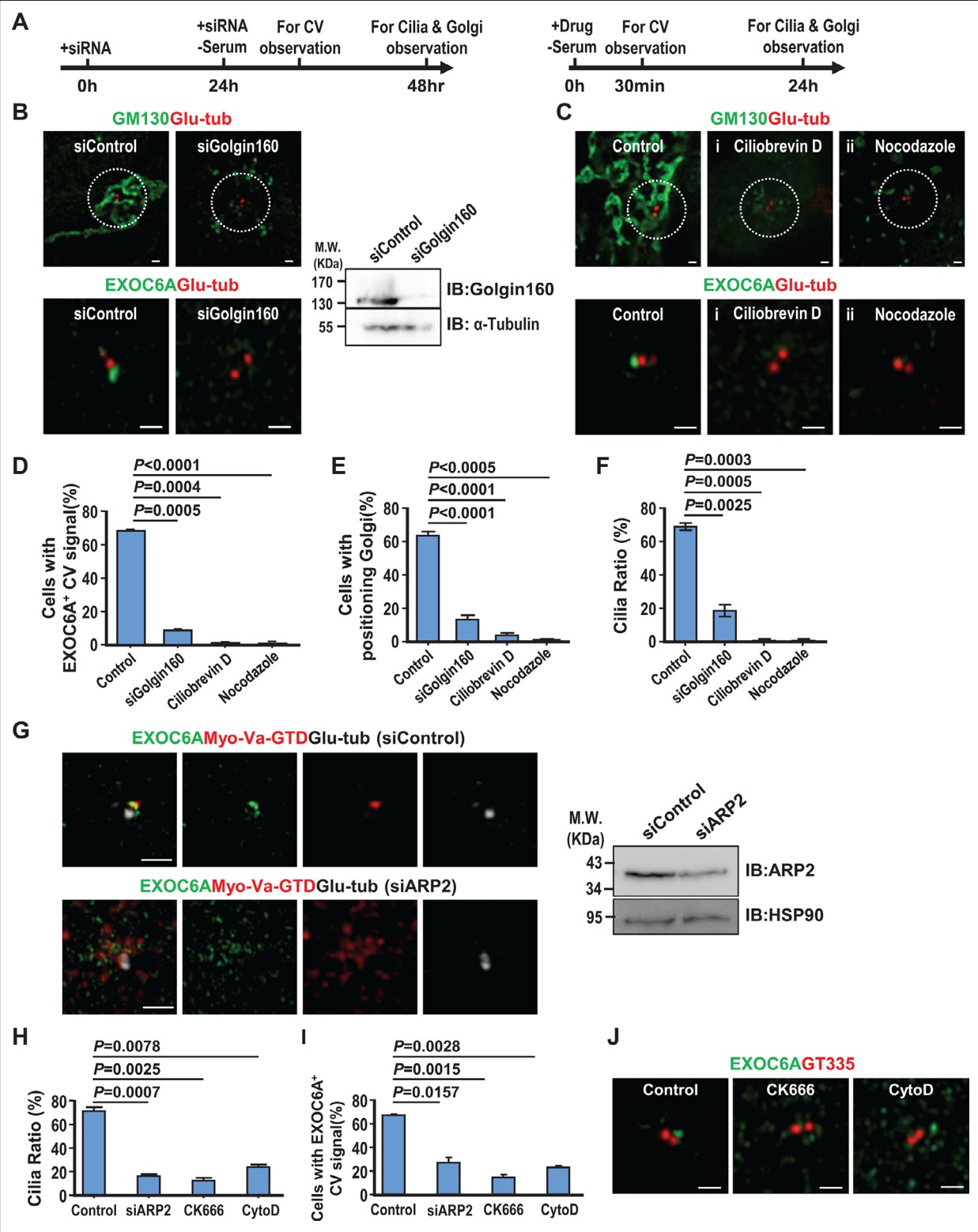

**Figure 5.** Transportation of EXOC6A-associated vesicles to the mother centriole occurs via a dynein-, actin-, and microtubule (MT)-dependent pathway. (**A**) Schematic showing the protocols of siRNA (left) or drug (right) treatment. (**B**) RPE1 cells were treated with siControl or siGolgin160 and analyzed via confocal fluorescence microscopy or immunoblotting using the indicated antibodies. (**C**) RPE1 cells were treated with 10 µM ciliobrevin D (**i**) or 10 µM nocodazole (**ii**) and analyzed via confocal fluorescence microscopy using the indicated antibodies. (**D–F**) Cells with EXOC6A-associated ciliary vesicles (CVs) (**D**), a positioned Golgi (**E**), or a cilia ratio (**F**) were analyzed via three-dimensional structured illumination microscopy (3D-SIM) and quantified. Cells

*Figure 5 continued on next page*

*Figure 5 continued*

with a positioned Golgi were defined as those showing all GM130-labeled Golgi signals concentrated within a 7 µm radius surrounding the glutamylated tubulin (Glu-tub)-labeled centrioles. (**G**) RPE1-based mCherry-myosin-Va-GTD-inducible cells were treated with siARP2 as described in A (left) and analyzed via confocal microscopy or immunoblotting using the indicated antibodies. (**H, I**) Quantifications of the cilia ratio (**H**) and EXOC6A-associated CVs (**I**) are shown. (**J**) RPE1 cells were treated with 200 µM CK666 or high-dose (10 µM) CytoD as described in A (right panel) and analyzed via confocal microscopy using the indicated antibodies. Quantification of the cilia ratio (**H**) and EXOC6A-associated CVs (**I**) is shown. * in B (right panel) indicates non-specific bands. Error bars represent mean ± s.d. from at least 3 independent experiments with 100 randomly selected cells. p-Value was determined with two-tailed Student's *t*-test. p<0.05 was considered statistically significant. NS, not significant. Scale bars are 1 µm.

The online version of this article includes the following source data and figure supplement(s) for figure 5:

**Source data 1.** PDF file containing original western blots for *Figure 5B and G*, indicating the relevant bands and treatments.

**Source data 2.** Original files for western blot analysis displayed in *Figure 5B and G*.

**Figure supplement 1.** Low doses of cytochalasin D (CytoD) promote cilia elongation, whereas higher concentrations (greater than 4 µM) inhibit ciliogenesis.

Since Myo-Va-mediated transport of PCVs to the mother centrioles relies on an Arp2/3-dependent branched actin network surrounding mother centrioles (*Wu et al., 2018*), we next treated RPE1 cells with siARP2 (*Figure 5G*). A reduced cilia number (*Figure 5G and H*) and reduced EXOC-6A-labeled CVs (*Figure 5G and I*) were observed in siARP2-treated cells. A similar result was also found in cells treated with 200 µM CK666 (a known specific Arp2/3 inhibitor) or high-dose (10 µM) cytochalasin D (CytoD), which has been reported to reduce the association of ARP2 with the branched actin network (*Figure 5H–J*). It has been reported that low concentrations (100 nM to 1 µM) of CytoD can promote ciliogenesis and increase cilia length (*Cao et al., 2023*; *Kim et al., 2015*; *Kim et al., 2010*; *Wu et al., 2018*), while high concentrations (10 µm) lead to a significant inhibition of cilia formation (*Wu et al., 2018*). To further investigate the dose-dependent effects of actin cytoskeleton rearrangement on ciliogenesis, we treated RPE1 cells with CytoD at concentrations ranging from 200 nM to 10 µM. Immunofluorescence staining of cilia using ARL13B (green) and Glu-tubulin (red) antibodies showed that low concentrations of CytoD (200 nM to 2 µM) resulted in significant cilia elongation, whereas higher doses (4 µM, 7 µM, and 10 µM) gradually inhibited cilia formation (*Figure 5—figure supplement 1*). Consistent with our previous findings (*Wu et al., 2018*), our data showed that cilia length and cilia ratio increased at low doses of CytoD (200 nM to 2 µM) but gradually decreased with increasing doses (7 µM and 10 µM, *Figure 5—figure supplement 1B and C*).

Taken together, our current studies reinforce our previous view (*Wu et al., 2018*) that low doses of CytoD treatment (200 nM to 2 µM) can stabilize and/or enhance the formation of Arp2-associated branched actin filaments. However, when cells were treated with high doses of CytoD (7 µM and 10 µM, *Figure 5—figure supplement 1*) or the specific Arp2/3 inhibitor, CK-666 (*Figure 5*), the association of Arp2 with the branched actin network was significantly reduced, thereby inhibiting ciliogenesis. In conclusion, our data suggest that Myo-Va carries EXOC6A vesicles to the DA of the mother centriole via a dynein-, MT-, and actin-dependent pathway.

## Loss of EXOC6A inhibits ciliogenesis, and some cells exhibit abnormal ciliary morphology when passing through the CV block

We found that EXOC6A signals are detected at CVs and the ciliary sheath (*Figures 1 and 2*). We also found that EXOC6A-associated vesicles are dynamically recruited to or excreted from CVs and/or the ciliary membrane (*Figure 3*). We next examined the effects of *EXOC6A* deletion on ciliogenesis. As shown in *Figure 6*, *EXOC6A* KO significantly inhibits cilia assembly, resulting in either no cilia or dotted CV signals after serum starvation (SS, 30 min to 48 hr). Upon serum starvation for 30 min, Myo-Va-GTD-labeled CVs were detected at the mother centriole in both wild-type (WT) control (*Figure 6A*) and *EXOC6A* KO cells (*Figure 6A'*). In control cells, the Myo-Va-GTD signal was mainly detected at the ciliary sheath membrane (*Figure 6A*, SS 2 hr) 2 hr after serum starvation, while the Myo-Va-GTD signals dissociated from the ciliary sheath membrane after longer starvation (SS 24 hr or SS 48 hr). Interestingly, most Myo-Va-GTD signals in *EXOC6A* KO cells were present as dotted CVs after longer starvation (SS 48 hr), suggesting that ciliogenesis is arrested at the CV stage (*Figure 6A'*).

We next used an antibody against ARL13B (a cilia shaft marker) to label the ciliary membrane and found that most *EXOC6A* KO cells exhibited no cilia (~45%, *Figure 6C*) or dotted ARL13B signals

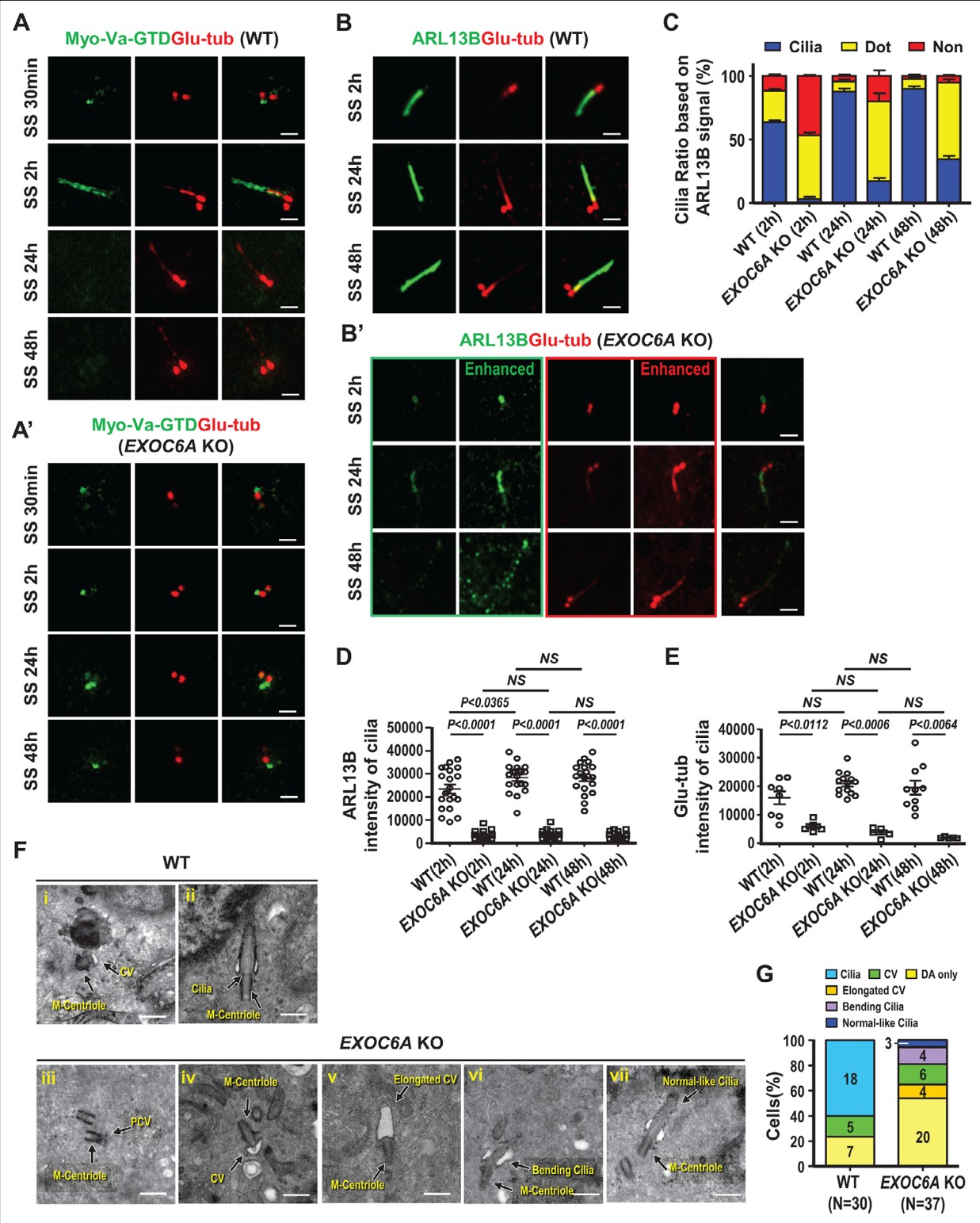

**Figure 6.** Loss of EXOC6A inhibits ciliogenesis, and some cells exhibit an abnormal ciliary morphology when passing through the ciliary vesicle (CV) block. (**A–C**) RPE1-based mCherry-Myo-Va-GTD-inducible wild-type (WT) (**A, B**) and *EXOC6A* knockout (KO) (**A', B'**) cells were treated with doxycycline (Dox) for 24 hr, serum-starved (SS) at different time points, and analyzed via confocal fluorescence microscopy with the indicated antibodies. mCherry-Myo-Va-GTD signal is artificially converted to green for discrimination. (**B**) RPE1 WT and *EXOC6A* KO cells were serum-starved at different time points and analyzed via confocal fluorescence microscopy with the indicated antibodies. For comparison, the intensities of ARL13B and glutamylated tubulin (Glu-tub) of *EXOC6A* KO cells with non-enhanced and enhanced signals are shown in B'. (**C**) Cilia ratio measured through ARL13B labeling (non-

*Figure 6 continued on next page*

*Figure 6 continued*

enhanced signal). (**D, E**) Quantitation of non-enhanced ARL13B (**D**) and non-enhanced Glu-tub intensity on cilia (**E**) of WT or *EXOC6A* KO cells is shown. (**F**) WT and *EXOC6A* KO cells were serum-starved for 72 hr. Morphologies of normal or abnormal cilia, including distal appendage (DA), CVs, and ciliary membranes, were examined via EM (**F**) and quantified (G; N is the number of cells examined). Error bars in C, D, and E represent mean ± s.d. from at least 3 independent experiments from 100 randomly selected cells. p-Value was determined with two-tailed Student's *t*-test. $p < 0.05$ was considered statistically significant. NS, not significant. Scale bars are 1 µm in A and B. Scale bars are 500 nm in F.

The online version of this article includes the following figure supplement(s) for figure 6:

**Figure supplement 1.** Rescue of ciliogenesis defects in EXOC6A knockout (KO) cells by GFP-EXOC6A re-expression.

(~50%) 2 hr after SS (*Figure 6B'*; *Figure 6C*, SS 2 hr), a pattern similar to Myo-Va staining (*Figure 6A'*). Unexpectedly, after prolonged serum starvation, we could still detect approximately ~15% (SS 24 hr) to ~30% (SS 48 hr) of *EXOC6A* KO cells carrying ARL13B-labeled cilia protruding from the basal body. However, these ARL13B-labeled cilia showed a low intensity of ARL13B and Glu-tub signals (*Figure 6D and E*) and the ARL13B-labeled cilia membrane exhibited a fragmented and punctuated staining pattern (enhanced in *Figure 6B'*, SS 48 hr in lower panel). Our findings suggest that EXOC6A may play a crucial role in the further maturation of CVs into the ciliary sheath/shaft membrane during ciliogenesis.

To examine whether EXOC6A is required for ciliogenesis and whether its function can be rescued, we analyzed cilia formation in WT, EXOC6A KO, and EXOC6A KO cells exogenously expressing GFP-EXOC6A. Immunofluorescence staining of cells with ARL13B and Glu-tubulin antibodies showed a significant reduction in the number of cilia in EXOC6A KO cells. Importantly, exogenous expression of GFP-EXOC6A in the KO cells resulted in a statistically significant recovery of the ciliated cell population (p=0.0041) and a concomitant reduction in stalled vesicles, confirming that this phenotype was caused by EXOC6A loss (*Figure 6—figure supplement 1*).

Electron microscopy was further used to investigate the cilia morphology in the *EXOC6A* KO cells after prolonged serum starvation (72 hr). To examine ciliary structure in *EXOC6A* KO cells in more detail, we used a 72 hr serum starvation protocol, which provided sufficient time for cilia assembly. Our results showed a perturbation in cilia assembly in *EXOC6A* KO cells (*Figure 6F and G*). When compared with the WT control (*Figure 6Fi and ii*), cilia in *EXOC6A* KO cells exhibited some ciliogenic defects, such that ciliogenesis was often arrested at the DA/PCV or CV stage (*Figure 6Fiii and iv*), and some exhibited morphologically curved cilia bending in the TZ region (*Figure 6Fvi*, 4/37 in *Figure 6G*). Intriguingly, a small proportion of KO cells produced an elongated large CV in the absence of an extended axoneme (*Figure 6Fv*, 4/37 in *Figure 6G*). Consistent with this finding, curved cilia (*Figure 6B'*, SS 24 hr, lower panel) and a low intensity of Glu-tub-labeled axoneme were observed in *EXOC6A* KO cells via confocal immunofluorescence microscopy (*Figure 6B'*, SS 48 hr, lower panel, *Figure 6E*). Nonetheless, we still observed 3 out of 37 morphologically normal-like cilia in *EXOC6A* KO cells (*Figure 6Fvii and G*). Taken together, our data suggest that depletion of EXOC6A primarily affects cells exhibiting no CV (showing only DA/PCV; *Figure 6Fiii and G*) and/or arrests cells at the CV stage, while once it passes through CV stage after prolonged serum starvation, it may interfere with subsequent cilia membrane formation at later stages of ciliogenesis.

## EXOC6A is required to recruit NPHP and MKS module components to the TZ, and GPR161 and BBS9 to the ciliary membrane

Our EM study revealed both bending cilia at the TZ (4/37, *Figure 6Gvi*) and morphologically normal-like cilia (3/37, *Figure 6Gvii*). Intriguingly, immunofluorescence analysis revealed that the ARL13B signal intensity was greatly reduced in the normal-like cilia membrane in *EXOC6A* KO cells (*Figure 6D*), suggesting a possible defect in the TZ of cilia. The TZ is a diffusion barrier at the ciliary membrane that acts as a gate to regulate the entry and exit of ciliary proteins required for signal transduction (*Jensen and Leroux, 2017*). Previous studies have identified three conserved protein modules (MKS, NPHP, and CEP290), composed of numerous protein complexes that cooperate in the assembly and gating function of the TZ (*Gonçalves and Pelletier, 2017*). CEP290, located at a proximal position near the mother centriole, acts as a hub that interacts with the NPHP and MKS module complexes (*Garcia-Gonzalo and Reiter, 2017*).

To verify the integrity of these abnormal cilia in *EXOC6A* KO cells, we examined the recruitment of several known TZ proteins, including CEP290, NPHP, and MKS module proteins. Our results showed

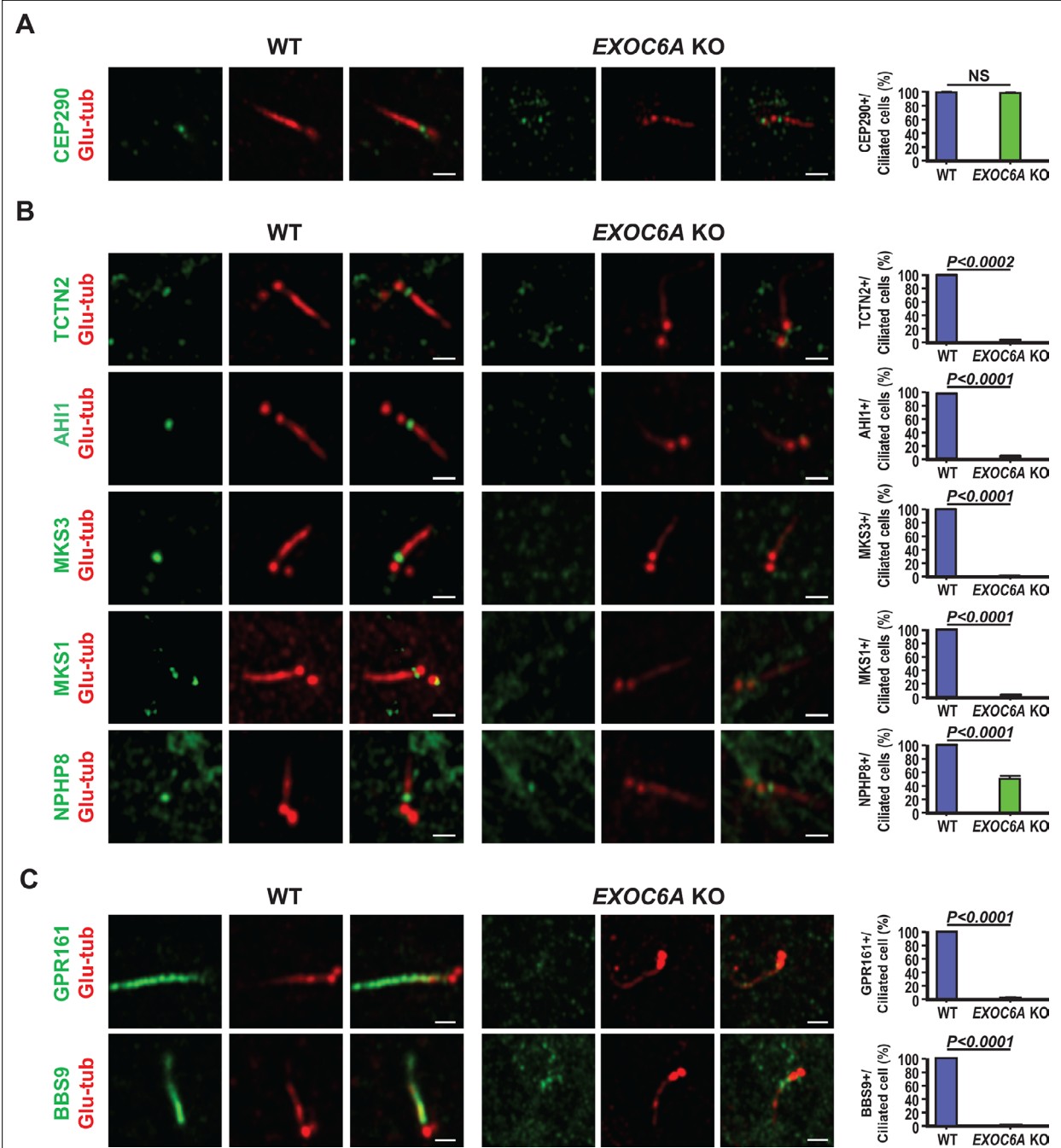

**Figure 7.** EXOC6A is required for the recruitment of a subset of transition zone (TZ) proteins and ciliary membrane proteins to the cilia. (**A**) Wild-type (WT) and *EXOC6A* knockout (KO) cells were fixed for 2 hr (**A**) or 24 hr (**B, C**) after serum starvation and analyzed via confocal fluorescence microscopy using indicated antibodies. Glutamylated tubulin (Glu-tub) staining labeled the centriole. Quantification of the fluorescence signals of various MKS proteins, NPHP8, and ciliary membrane proteins (GRP161 and BBS9) is shown in the right panel. Error bars represent the mean ± s.d. from at least 3 independent experiments with 100 randomly selected cells per experiment. p-Value was determined with two-tailed Student's *t*-test. p<0.05 was considered statistically significant. NS, not significant. Scale bars, 1 μm.

that the recruitment of CEP290 to the ciliary base was normal in all examined ciliated cells, indicating that EXOC6A deletion did not affect CEP290 targeting to the TZ (*Figure 7A*). However, the recruitment of several MKS complex proteins (TCTN2, AHI1, MKS3, and MKS1) to the basal body (mother centriole) was severely impaired in *EXOC6A* KO ciliated cells (*Figure 7B*), while NPHP8 recruitment was only partially affected (*Figure 7B*). Collectively, our findings suggest that EXOC6A is not required

for CEP290 targeting to the TZ but plays an essential role in the recruitment of several NPHP and MKS module proteins to the basal body, which potentially affects the gating function of the TZ in *EXOC6A* KO cells.

We next investigate whether the EXOC6A deficiency affects the transport of ciliary membrane proteins, we examined the localization of GPR161 and BBS9 in *EXOC6A* KO cells. GPR161 is a G-protein-coupled receptor that negatively regulates the Sonic hedgehog signaling pathway (*Mukhopadhyay et al., 2013*), while BBS9 is a core component of the BBSome complex and is essential for cilia homeostasis (*Wingfield et al., 2018*). Our results showed that both GRP161 and BBS9 were properly recruited to the cilia in WT cells (*Figure 7C*). However, in *EXOC6A* KO cells, the entry of these proteins into the ciliary compartment was completely blocked (*Figure 7C*). Our findings suggest that EXOC6A is essential for the recruitment of a subset of TZ proteins and membrane proteins to the cilia during ciliogenesis.

## Discussion

Cilia assembly is a complex, multistep process involving the accumulation and attachment of PCVs at the DA, CV formation, and the subsequent extension of CVs into the ciliary sheath, alongside the formation of the ciliary shaft, ultimately leading to the mature cilia structure (*Breslow and Holland, 2019*; *Long and Huang, 2019*; *Nachury et al., 2010*). Currently, our understanding of the molecular mechanisms governing these steps remains incomplete. Exocytosis is a process where intracellular secretory vesicles fuse with the plasma membrane, aiding in delivering integral membrane proteins to the cell surface and releasing materials into the extracellular space (*Martin-Urdiroz et al., 2016*). Human EXOC6A exhibits a high degree of similarity to the product of the *Saccharomyces cerevisiae SEC15* gene. In yeast, SEC15 is essential for the movement of vesicles from the Golgi apparatus to the cell surface (*Bowser and Novick, 1991*). Previous studies have indicated that SEC15 interacts with Myo2p (the yeast homolog of human Myo-Va) and plays a role in vesicle exocytosis (*Donovan and Bretscher, 2012*; *Donovan and Bretscher, 2015*). Recently, exocyst was found to mediate the recycling of internal cilia, and depletion of EXOC6A/Sec15a, but not other exocyst components, leads to a reduction in ciliogenesis (*Rivera-Molina et al., 2021*). This finding suggests that EXOC6A may have a unique function for ciliogenesis. However, the relationship and underlying mechanism between EXOC6A and Myo-Va in the context of ciliogenesis remain unclear.

In this study, we elucidated the role of the exocyst component EXOC6A in ciliogenesis and produced evidence for the model by which Myo-Va mediates trafficking of EXOC6A vesicles to the DA or existing CV of mother centriole in the early stages and to the ciliary membrane in the later stages of ciliogenesis (*Figure 8*). However, further investigation will be required to fully establish the proposed hierarchy and molecular details. In this model, we found that a subset of EXOC6A-positive signals colocalized with Myo-Va-labeled signals during the early phase of ciliogenesis, and that they exhibited a high correlation during clustering-to-centriole progression after initiating ciliogenesis (*Figure 2—figure supplement 1A–D*). Intriguingly, not all EXOC6A-labeled signals were positive for Myo-Va (*Figure 2A*), suggesting the presence of at least two populations of Myo-Va-labeled vesicles in cells: one that is EXOC6A+/Myo-Va+ and another that is EXOC6A-/Myo-Va+ (*Figures 2 and 8*).

In this model, ciliogenesis begins shortly after serum-starvation-induced quiescence. Initially, Myo-Va mediates the transport of PCVs (EXOC6A-/Myo-Va+ and EXOC6A+/Myo-Va+ vesicles) to the DA of the mother centriole, followed by the formation of PCVs to a CV via EHD1 (*Lu et al., 2015*; *Wu et al., 2018*). However, EXOC6A is not absolutely required for CV formation, as *EXOC6A* KO did not interfere with the formation of Myo-Va- or EHD1-labeled CV (*Figure 4B and D*). Once the CV is formed, the transport of EXOC6A vesicles to the existing CV in the early stages of ciliogenesis is still mediated by Myo-Va (*Figure 4A*). The above process involves a two-step transportation mechanism: initially, dynein mediates the movement of Myo-Va-associated vesicles (either EXOC6A-positive or EXOC6A-negative) toward the pericentrosomal region along MTs (*Figure 5B–F*). Subsequently, Myo-Va takes over and drives these vesicles (EXOC6A-positive or -negative) from the pericentrosomal region to the DAs of the mother centriole. This process is accomplished through the utilization of the ARP2/3-associated branched actin network (*Figure 5G–J*). Notably, both of these Myo-Va-labeled vesicles require the function of EHD1 to form a CV (*Figure 4D*; *Wu et al., 2018*). Our findings that EHD1 depletion (*Figure 4D*) and *Myo-Va* KO (*Figure 4A*) severely impaired the formation of

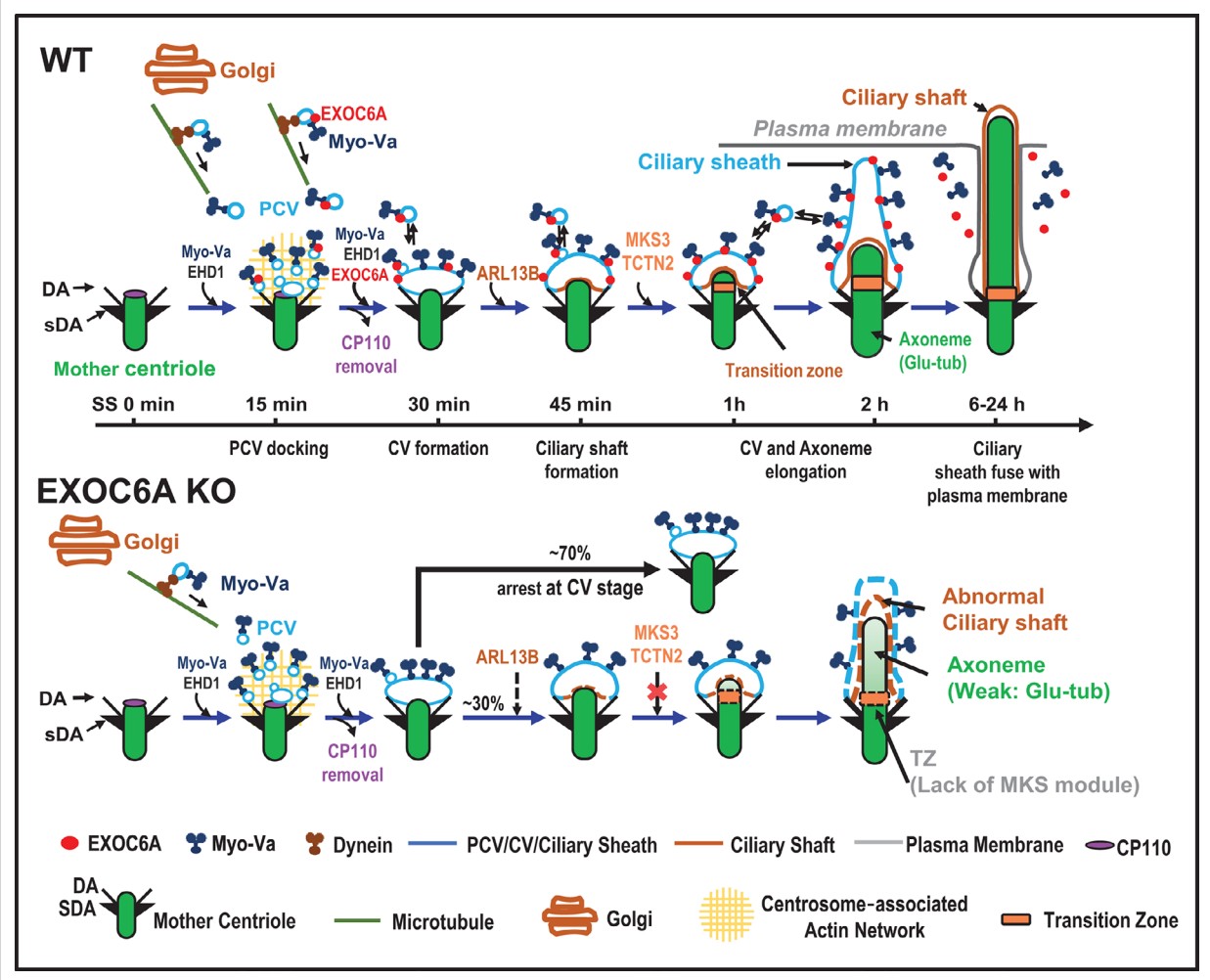

**Figure 8.** Ciliogenesis model and comparison between wild-type (WT) and *EXOC6A* knockout (KO) cells. EXOC6A is located at the PCV/CV/ciliary sheath, and myosin-Va (Myo-Va) mediated the transportation of EXOC6A vesicles to the mother centriole via a dynein-, microtubule-, and actin-dependent pathway. EXOC6A-associated vesicles are continuously recruited and fused or excreted from the CVs or ciliary membrane. Depletion of EXOC6A impairs ciliogenesis, and most cells are arrested at the CV stage. There is a lack of MKS module proteins of the TZ in cilia with defects in *EXOC6A* KO cells. PCV, preciliary vesicle; CV, ciliary vesicle; DA, distal appendage; sDA, subdistal appendage.

EXOC6A-positive CVs at the mother centriole suggest that Myo-Va and EHD1 play a role in promoting the formation of EXOC6-positive CVs during the early stages of ciliogenesis.

Although EXOC6A is not absolutely required for the initial CV formation (*Figure 4B and D*), our data revealed another role of EXOC6A in the later stages of ciliogenesis. Our FRAP analysis and live-cell imaging showed that a subset of EXOC6A-labeled vesicles displayed high dynamism throughout ciliogenesis, continually being recruited to, fusing into, or exited from existing CVs in the early stages and/or the ciliary membrane (or ciliary pocket) in the later stages of ciliogenesis (*Figure 3*). These EXOC6A-positive vesicles, likely derived from the Golgi or recycling endocytic vesicles, are transported to the ciliary membrane (or ciliary pocket) and/or fused with/exited from the ciliary membrane in the later stages of ciliogenesis. The exocyst has been implicated in the transport of vesicles carrying ciliary proteins to the cilia (*Wu and Guo, 2015*). Thus, our findings suggested a potential role of EXOC6A vesicles, which may supply or remove proteins or lipid moieties from the CVs and/or the ciliary membrane (or ciliary pocket) during ciliogenesis.

Additionally, our results showed that *EXOC6A* KO led to the majority of the cells being arrested at the CV stage or exhibiting no or few cilia (*Figure 6C*); only a small proportion of cilia were able to elongate, exhibiting an abnormal ciliary morphology (*Figure 6B and F*). Electron microscopic analysis of these abnormal cilia revealed elongated CV (lacked axoneme, *Figure 6Fv*), curved cilia bending at

the TZ region (*Figure 6Fvi*), or normal-like cilia (*Figure 6Bvii*, whose structure and membrane composition need to be further characterized). Further confocal immunofluorescence analysis revealed a curved ciliary morphology (*Figure 6B'*, SS 24hr) and impaired recruitment of several known TZ proteins, including TCTN2, AHI1, MKS3, and MKS1, to the basal body in *EXOC6A* KO cells (*Figure 7*). Taken together, our findings suggest that EXOC6A is required for the early and the later stages of ciliogenesis as providing or removing ciliary components (proteins or membrane moieties) to the existing CV or ciliary membrane, which highlights its multifaceted involvement in the process of ciliary membrane formation.

Furthermore, the Rab11-Rabin8-Rab8 signaling pathway, which involves the recruitment of Rabin8 (a GDP-GTP exchange factor for Rab8) and subsequent activation of Rab8, has been implicated in ciliary membrane assembly (*Knödler et al., 2010*; *Nachury et al., 2007*; *Westlake et al., 2011*; *Yoshimura et al., 2007*). Recently, RAB34, located at the ciliary sheath (*Ganga et al., 2021*; *Stuck et al., 2021*), was also reported to be required for ciliogenesis (*Xu et al., 2018*). The relationship between EXOC6A, Rab11 signaling, and RAB34 in ciliogenesis is not clear. Future experiments dissecting the underlying mechanisms between Rab11 signaling, RAB34, and EXOC6A could provide a more complete picture of ciliary membrane assembly during ciliogenesis.

Finally, ciliopathy-associated genes have been demonstrated to impact various stages of mouse cortical development (*Guo et al., 2015*), and severe human ciliopathies are linked to developmental abnormalities in the forebrain (*Andreu-Cervera et al., 2021*). Mutations or deletions in genes encoding exocyst proteins, including *EXOC2*, *EXOC4*, *EXOC7*, and *EXOC8*, have been associated with ciliopathies and neural developmental disorders (*Coulter et al., 2020*; *Dixon-Salazar et al., 2012*; *Shaheen et al., 2013*; *Turkyilmaz et al., 2021*; *Van Bergen et al., 2020*). Interestingly, deletions involving the 5' portion of the *EXOC6A* gene and two adjacent cytochrome P450 genes (*CYP26A1* and *CYP26C1*) are associated with autosomal-dominant nonsyndromic optic aplasia (ONA), an extremely rare disorder that causes unilateral or bilateral blindness in the affected eye (*Meire et al., 2011*). The molecular basis remains unclear. Further studies are needed to explore the specific roles of various types of CVs, the underlying mechanisms of their trafficking/fusion, and their relevance to ciliopathies and neural developmental disorders.

In summary, we have identified a novel role for EXOC6A in ciliogenesis and determined that it is essential for both early and late stages of this process. EXOC6A vesicles may transport proteins or lipid moieties to and from the CV (early phase) and the ciliary pocket/ciliary membrane (late phase) during ciliogenesis. However, EXOC6A is not absolutely necessary for the initial CV formation; its function may be to help assemble the complete TZ and promote the maturation of existing CVs into the ciliary sheath/shaft membrane. Furthermore, transportation of EXOC6A vesicles to the mother centriole is mediated by Myo-Va and occurs via a dynein-, MT-, and Arp2/3-dependent pathway. Our study provides new insights into the function of EXOC6A in cilia membrane assembly during ciliogenesis. Future studies are needed to explore the specific roles of different types of CVs, the underlying mechanisms of their trafficking and fusion, and their relevance to ciliopathies and neural developmental disorders.

## Methods

### Plasmids and antibodies

cDNAs encoding full-length EXOC6A were obtained via RT-PCR using total RNAs from human HEK293T cells and subcloned in-frame into pcDNA4/TO/myc-His-A (Invitrogen) or pLVX-Tight-Puro (BD Biosciences Clontech) GFP expression vectors. GFP- and mCherry-myosin-Va-GTD constructs have been previously described (*Wu et al., 2018*). The sequences of all constructed plasmids were confirmed.

Commercial antibodies used were anti-Sec15 (clone [15S2G6], 1/100 for IF, Kerafast, Cat# ED2003), anti-EXOC6 (1/100 for IF and 1/2000 for WB, Novus Cat# NBP1-85031), anti-EXOC5 (1/2000 for WB, Proteintech, Cat# 17593-1-AP), anti-EXOC7 (1/2000 for WB, Proteintech, Cat# 12014-1-AP), anti-polyglutamylated tubulin (Glu-tub, GT335, 1/100 for IF, AdipoGen, Cat# AG-20B-0020-C100), anti-CEP164 (1/500 for IF, Novus, Cat# NBP1-81445), anti-GFP (Abcam, Cat# ab13970 and Clontech, Cat# 632381), anti-ARL13B (1/600 for IF, Proteintech, Cat# 17711-1-AP), anti-myosin-Va (1/500 for IF and 1/5000 for WB, Novus, Cat# NBP1-92156), anti-EHD1 (1/500

for IF and 1/5000 for WB, Novus, Cat# NBP2-56035), anti-GM130 (1/1000 for IF, Abcam, Cat# AB52649), anti-CP110 (1/600 for IF, Proteintech, Cat# 12780-1-AP), anti-Golgin160 (GOLGA3) (1/2000 for WB, Novus Cat# NBP1-91952), anti-Arp2 (1/1000 for WB, Abcam Cat# ab47654), anti-TMEM67 (MKS3, 1/400 for IF, Proteintech Cat# 13975-1-AP), anti-TCTN2 (1/400 for IF, Protein-tech Cat# 17053-1-AP), anti-AHI1 (1/400 for IF, Proteintech Cat# 22045-1-AP), anti-MKS1 (1/400 for IF, Proteintech Cat# 16206-1-AP), anti-BBS9 (1/400 for IF, Proteintech Cat# 14460-1-AP), and anti-GPR161 (1/400 for IF, Proteintech Cat# 13398-1-AP). The secondary antibodies were Alexa 488/568/647-conjugated anti-mouse or rabbit (Invitrogen). The rabbit polyclonal antibody against CEP120 (residues 639–986) was as described in our previous paper (*Lin et al., 2013*; *Tsai et al., 2019*).

## Cell culture and cell lines

Human retinal pigment epithelial cell line RPE-1 (also known as hTERT RPE-1) had been previously used (*Wu et al., 2018*) and validated by our institute's DNA Sequencing Core Facility using STR profiling analysis. RPE-1 cells were maintained in DMEM/F12 (1:1) supplemented with 10% FBS. Cells were transfected with various cDNA constructs using TransIT-LT1 Transfection Reagent (Mirus). All cell lines were tested for mycoplasma contamination and found negative.

The gRNA expression plasmids were obtained from Addgene (Addgene plasmid # 48138; http://n2t.net/addgene:4813), having been generated by inserting annealed primers into the gRNA cloning vector pSpCas9-2A-GFP (PX458) (*Ran et al., 2013*). The targeting sequences for the EXOC6A gRNA (gRNA1: 5'- TTGTATCCGTAATCATGACA -3',gRNA2: 5'- AAGTTACTGATACCAACCGA -3' and gRNA3: 5'- ACAAGTTTAACTCACCAGGAAGG -3') primer pair purchased from Mission Biotech were annealed and cloned into PX458. For transfection, 2.5 µg gRNA (PX458 plasmid) was transfected with LT-1 reagent (Mirus) into RPE1 cells, according to the manufacturer's instructions. The KO-1 clone was generated using a single-guide RNA strategy (*Figure 4—figure supplement 1Aa*). The PX458 vector encoding gRNA1, targeting exon 2 of the EXOC6A gene, was transfected into RPE1 cells. The KO-2 independent clone was generated using a dual-guide RNA strategy to induce a wider range of deletion (*Figure 4—figure supplement 1Ab*). Vectors encoding gRNA2 and gRNA3, targeting the flanking regions of exon 3 to exon 4, were co-transfected into cells. The transfected cells were sorted into single cells (96-well plates) using a cell sorter (FACSAria, BD Biosciences) based on GFP signal and then amplified. To confirm the depletion of EXOC6A, genomic DNAs isolated from *EXOC6A* KO colonies were subjected to PCR using the following primers: and 5'- ACATCTCCTGAGCCTCATACC -3' and 5'- GCTTCAGAAAAAGAGAATACTCCT –3' for *EXOC6A* KO-1 (gRNA1) and 5'- TTGGGTCA GTGATTTGAATTG -3' and 5'- CCAAATAATCTGTAATTCCCATA -3' for *EXOC6A* KO-2 (gRNAs 2 and 3). The sequences of PCR products were then confirmed as shown in *Figure 4—figure supplement 1*. Based on sequence analysis, a schematic diagram of the predicted amino acid sequence is shown in *Figure 4—figure supplement 1B*. Also, the protein expression of expanded colonies was confirmed by western blotting (*Figure 4—figure supplement 1C*) and IF (*Figure 4—figure supplement 1E and F*).

RPE1-based Dox-inducible cell lines were as described previously (*Wu et al., 2018*). To obtain RPE1 cell lines that inducibly expressed GFP-EXOC6A, GFP-Myo-Va-GTD, and mCherry-Myo-Va-GTD, lentiviruses generated using the target cDNAs in the pLVX-tight-puro vector were used to infect RPE1 Tet-On cells (stably expressing rtTA, Clontech). The infected cells were sterile-sorted using a cell sorter (FACSAria, BD Biosciences) for GFP or mCherry, and the positive cells were selected and expanded as inducible cell lines.

The siRNAs and non-targeting siRNA control were transfected into cells using Lipofectamine RNAiMAX (Invitrogen), according to the manufacturer's protocol. To increase knockdown efficiency, two rounds of silencing were used before the experiments. To ensure the specificity of gene knock-down and minimize off-target effects, we used siRNA sequences that have been validated in our previous studies (*Wu et al., 2018*). The efficiency of gene depletion was confirmed by western blot-ting. The siRNA sequences for siEHD1 (5'- GGAGAGAUCUACCAGAAGA-3'), siGolgin-160 (5'-CAGC CUCCUUGGCCGCGAGGGCCUC-3'), and siARP2 (5'-CCAGCUUUGGUUGGAAGACCUAUUA-3') were purchased from Invitrogen.

## Immunoprecipitation and western blotting

RPE1 cells were lysed in RIPA buffer (50 mM Tris-HCl, pH 8.0, 150 mM NaCl, 1% NP-40, 0.5% sodium deoxycholate, 20 mM β-glycerophosphate, 20 mM NaF, 1 mM $Na_3VO_4$) supplemented with protease inhibitors (1 mg/mL leupeptin, 1 mg/mL pepstatin, and 1 mg/mL aprotinin) for 1 hr at 4°C. Lysates were clarified by centrifugation for 10 min at 4°C, and the supernatants were immunoprecipitated with the indicated antibodies for 2 hr at 4°C. Protein G Sepharose beads were subsequently added and incubated overnight at 4°C with gentle rotation. The beads were then washed three times with RIPA buffer. Bound proteins were eluted by boiling in SDS sample buffer, separated by SDS-PAGE, and analyzed by western blotting with the appropriate antibodies.

## Immunofluorescence confocal microscopy

Immunofluorescence confocal microscopy was carried out as previously described (*Wu et al., 2018*). Briefly, cells were grown on coverslips and fixed in methanol at −20°C for 20 min. The cells were then blocked with 10% normal goat serum in PBST. After blocking, the cells were incubated with primary antibodies, washed with PBST, and then incubated with Alexa Fluor 488-, 568-, or 647-conjugated secondary antibodies (1/500; Invitrogen). The samples were visualized using an LSM880 Airyscan system (Carl Zeiss). FRAP experiments were performed on live RPE1 cells co-expressing GFP-EXOC6A and mCherry-Myo-Va-GTD using a Zeiss LSM 880 confocal microscope. Time-lapse imaging was conducted with a time interval of 20 s. At each time point, Z-stacks consisting of five optical sections were captured to cover the entire structure. Following the acquisition of five pre-bleach baseline images, the GFP-EXOC6A and mCherry-Myo-Va-GTD signals within a defined region of interest were photobleached using the 405 nm laser at 100% intensity for 100 iterations. The fluorescence recovery was monitored over time, and intensity changes were quantified from maximum intensity projections of the Z-stacks using ZEN software.

## Super-resolution microscopy (3D-SIM)

The cells were fixed and stained as described (see above), and multi-colored beads (100 nm; Invitrogen) were added before mounting. Sixteen-bit super-resolution images with five phases and five rotations were obtained using a Zeiss ELYRA PS.1 LSM780 system or a Zeiss ELYRA 7 system (Carl Zeiss). We used a plan-apochromatic 63×/1.4 NA oil-immersion M27 objective in combination with the default grid (stripe sim in ELYRA PS.1 and Lattice sim plus SIM square). We used the 'optimized' setting for the z-stack interval. The raw images were reconstructed using ZEN Black (Carl Zeiss) under the default parameters. The reconstructed images were corrected with signals of beads for chromatic aberration in the *x*, *y*, and *z* directions, and the images were aligned accordingly with ZEN Black. We followed the standard guidelines (*Demmerle et al., 2017*) and used the SIM-Check software (*Ball et al., 2015*) to verify the raw, reconstructed, and calibrated data. To visualize centriole and axoneme structures during live-cell imaging, we employed SiR-tubulin, a fluorogenic probe that binds to endogenous MTs (spirochrome).

## Ultrastructure expansion microscopy

U-ExM was used to reveal the fine cellular architecture of the cells. Cells were processed for this U-ExM as previously described (*Gambarotto et al., 2019*). In brief, cells were fixed with methanol or 4% PFA based on our target. First, cells on coverslips were incubated in a PBS solution containing 0.7% formaldehyde (FA; 36.5–38%; F8775; Sigma-Aldrich) plus 0.15% acrylamide (AA; 40%; A4058; Sigma-Aldrich) for 5 hr at 37°C. Next, the cells on coverslips were incubated in U-ExM monomer solution, composed of 19% (wt/wt) sodium acrylate (SA; 97–99%; 408220; Sigma-Aldrich), 10% (wt/wt) AA, and 0.1% (wt/wt) *N,N′*-methylenebisacrylamide (BIS; 2%; M1533; Sigma-Aldrich) at 37°C for 1 hr in the dark for gelation. We transferred coverslips with gel to six-well plates, added 1 mL denaturing buffer (200 mM SDS, 200 mM NaCl, and 50 mM Tris in ultrapure water, pH 9), and shook for 15 min at room temperature (RT). We then separated gel from coverslips and transferred the gel to 1.5 mL centrifuge tubes with fresh denaturing buffer, then incubated at 95°C for 30 min. After denaturation, the gels were placed in beakers filled with ddH$_2$O overnight for expansion. For antibody staining, the gels were PBS washed and then incubated with primary antibodies for 1 or 2 nights at RT (depending on the antibodies). The gels were washed in PBST (PBS plus 0.1% Tween 20) and incubated with a secondary

antibody for 3 hr at 37°C. The gels expanded by around 4× after this treatment. The samples were then visualized using an LSM880 Airyscan system (Carl Zeiss).

### TEM and CLEM

Cells were processed for transmission electron microscopy (TEM) and CLEM as previously described (*Reddick and Alto, 2012*). Briefly, RPE1 and RPE1-based *EXOC6A* KO cells were grown on Aclar film (Electron Microscopy Sciences) and fixed in 2% glutaraldehyde (GA) containing 1% tannic acid in 0.1 M cacodylate buffer. The fixed cells were post-fixed with 1% $OsO_4$ in 0.1 M cacodylate buffer at RT for 30 min. Then, the cells were further stained with 1% uranyl acetate (UA) at RT for 1 hr, dehydrated in a graded series of ethanol, infiltrated, and embedded in Spurr's resin. Ultrathin sections (70 nm) were stained with 4% UA and Reynold's lead citrate for 10 min and examined with an electron microscope (Tecnai G2 Spirit TWIN, FEI).

For CLEM, RPE1-based GFP-EXOC6A-inducible cells were grown on laser-etched glass gridded coverslips affixed to the bottom of live-cell dishes (MatTek). Cells were treated with Dox for 24 hr and serum-starved for 2 or 24 hr. The cells were then fixed with 2% GA plus 1% tannic acid in 0.1 M cacodylate buffer, and the GFP-EXOC6A signals were imaged via 3D-SIM using a Zeiss ELYRA system (Carl Zeiss). The cells were then embedded in EPON resin. After EPON polymerization, resin blocks were detached from the glass dishes. Using the grid patterns imprinted in the resin, serial thin sections (70 nm) of the squares of interest were cut. The sections were stained with 4% UA and Reynold's lead citrate for 10 min, and the cells were examined with an electron microscope (Tecnai G2 Spirit TWIN, FEI). All EM images were processed using Gatan Digital Micrograph software (Gatan).

### Statistics and reproducibility

Statistical analyses were performed using GraphPad Prism. Results are presented as mean ± standard deviation (s.d.), as specified in the figure legends. Sample size and the number of repeated experiments are also described in the legends. Statistical differences between two datasets were analyzed using the two-tailed paired Student's *t*-test. The precise p-values are shown in the figures. $p < 0.05$ was considered statistically significant. All experiments were repeated at least three times.

## Acknowledgements

The authors acknowledge support from the Flow Cytometry Core Facility (IBMS, AS-CFII-111-212) for the cell-sorting service, the DNA Sequencing Core Facility (IBMS, AS-CFII-113-A12) for the DNA sequencing analysis, the Light Microscopy Core Facility (IBMS and IMB), and the EM core facilities (IMB and ICOB) of Academia Sinica. This work was supported by grants from Academia Sinica (AS-IA-109-L04) and the National Science and Technology Council, Taiwan (NSTC 112-2320-B-001-002; NSTC 112-2326-B-001-010) to TKT; and from the National Institutes of Health R01AI184984 to CTW.

## Additional information

### Funding

| Funder | Grant reference number | Author |
| --- | --- | --- |
| Academia Sinica | AS-IA-109-L04 | Tang K Tang |
| National Science and Technology Council | NSTC 112-2320-B001-002 | Tang K Tang |
| National Institutes of Health | R01A1184984 | Chien-Ting Wu |
| National Science and Technology Council | NSTC 112-2326-B-001-010 | Tang K Tang |

The funders had no role in study design, data collection and interpretation, or the decision to submit the work for publication.

## Author contributions
Te-Lin Lin, Conceptualization, Resources, Investigation, Methodology, Writing – original draft, Writing – review and editing; Chien-Ting Wu, Investigation, Methodology, Writing – review and editing; Tang K Tang, Conceptualization, Resources, Supervision, Funding acquisition, Methodology, Writing – original draft, Writing – review and editing

## Author ORCIDs
Tang K Tang ⬚ https://orcid.org/0000-0002-3660-0695

Reviewer #2 (Public review): https://doi.org/10.7554/eLife.108271.3.sa1
Reviewer #3 (Public review): https://doi.org/10.7554/eLife.108271.3.sa2
Author response https://doi.org/10.7554/eLife.108271.3.sa3

# Additional files

## Supplementary files
MDAR checklist

## Data availability
All data generated or analyzed during this study are included in the manuscript and supporting files.

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
