## [Editor Report · eLife Assessment]

This **important** study elucidates the role of the exocyst component EXOC6A at distinct stages of ciliogenesis, which advances our understanding of ciliary membrane remodeling and cilium formation. The authors provide **compelling** evidence through high quality light and electron microscopic imaging, and careful analysis of knockout cell lines, that EXOC6A interacts with myosin-Va and is dynamically recruited via dynein-, microtubule-, and actin-dependent mechanisms, to support proper formation of the ciliary membrane. The study will be of interest to cell biologists and other researchers interested in vesicular trafficking, organellar membrane dynamics, and ciliogenesis.

---

## [Referee Report · Reviewer #2 (Public review)]

Summary:

The molecular mechanisms underlying ciliogenesis are not well understood. Previously, work from the same group (Wu et al., 2018) identified myosin-Va as an important protein in transporting preciliary vesicles to the mother vesicles, allowing for initiation of ciliogenesis. The exocyst complex has previously been implicated in ciliogenesis and protein trafficking to cilia. Here, Lin et al. investigate the role of exocyst complex protein EXOC6A in cilia formation. The authors find that EXOC6A localizes to preciliary vesicles, ciliary vesicles, and the ciliary sheath. EXOC6A colocalizes with Myo-Va in the ciliary vesicle and the ciliary sheath, and both proteins are removed from fully assembled cilia. EXOC6A is not required for Myo-Va localization, but Myo-VA and EHD1 are required for EXOC6A to localize in ciliary vesicles. The authors propose that EXOC6A vesicles continually remodel the cilium: FRAP analysis demonstrates that EXOC6A is a dynamic protein, and live imaging shows that EXOC6A fuses with and buds off from the ciliary membrane. Loss of EXOC6A reduces, but does not eliminate, the number of cilia formed in cells. Any cilia that are still present are structurally abnormal, with either bent morphologies or transition zone defects. Overall, the analyses and imaging are well done, and the conclusions are well supported by the data. The work will be of interest to cell biologists, especially those interested in centrosomes and cilia.

Strengths:

The TEM micrographs are of excellent quality. The quality of the imaging overall is very good, especially considering that these are dynamic processes occurring in a small region of the cell. The data analysis is well done and the quantifications are very helpful. The manuscript is well-written and the final figure is especially helpful in understanding the model.

The manuscript has greatly improved after revision. In particular, testing GPR161 and BBS9 localization is helpful evidence to demonstrate that transition zone function is disrupted when EXOC6A is lost. The generation of a second knockout clone and tests of antibody specificity are also great additions.

Weaknesses:

None

---

## [Referee Report · Reviewer #3 (Public review)]

Summary:

Lin et al report on the dynamic localization of EXOC6A and Myo-Va at pre-ciliary vesicles, ciliary vesicles, and ciliary sheath membrane during ciliogenesis using three-dimensional structured illumination microscopy and ultrastructure expansion microscopy. The authors further confirm the interaction of EXOC6A and Myo-Va by co-immunoprecipitation experiments and demonstrated the requirement of EHD1 for the EXOC6A-labeled ciliary vesicles formation. Additional experiments using gene-silencing by siRNA and pharmacological tools identified the involvement of dynein-, microtubule-, and actin in the transport mechanism of EXOC6A-labeled vesicles to the centriole, as they have previously reported for Myo-Va. Notably, loss of EXOC6A severely disrupts ciliogenesis, with the majority of cells becoming arrested at the ciliary vesicle (CV) stage, highlighting the involvement of EXOC6A at later stages of ciliogenesis. As the authors observe dynamic EXOC6A-positive vesicle release and fusion with the ciliary sheath, this suggests a role in membrane and potentially membrane protein delivery to the growing cilium past the ciliary vesicle stage. While CEP290 localization at the forming cilium appears normal the recruitment of other transition zone components, exemplified by several MKS and NPHP module components, was also impaired in EXOC6A-deficient cells.

Strengths:

- By applying different microscopy approaches, the study provides deeper insight into the spatial and temporal localization of EXOC6A and Myo-Va during ciliogenesis.

- The combination of complementary siRNA and pharmacological tools targeting different components strengthens the conclusions.

- This study reveals a new function of EXOC6A in delivering membrane and membrane proteins during ciliogenesis, both to the ciliary vesicle as well as to the ciliary sheath.

- The overall data quality is high. The investigation of EXOC6A at different time points during ciliogenesis is well schematized and explained.

- The authors confirmed central antibody reagents used in this study and validated key experiments by using two independent knockout clones (for which sequencing information was provided).

Weaknesses:

- The precise molecular function of EXOC6A remains open, as the presented data suggests no involvement of other exocyst components.

Taken together, the authors achieved their goal to elucidate the role of EXOC6A in ciliogenesis, demonstrating its involvement in vesicle trafficking and membrane remodeling in both early and late stages of ciliogenesis. Their findings are supported by experimental evidence. This work is likely to have an impact on the field by expanding our understanding of the molecular machinery underlying cilia biogenesis, particularly the coordination between exocyst components and cytoskeletal transport systems. The methods and data presented offer valuable tools for dissecting vesicle dynamics and cilium formation, providing a foundation for future research into ciliary dysfunction and related diseases. By connecting vesicle trafficking to structural maturation of an organelle, the study adds important context to the broader description of cellular architecture and organelle biogenesis.

Comments on revisions:

We very much appreciate the extra work you put into improving your manuscript and want to congratulate you on your important discovery. We encourage you to keep up the good work!

---

## [Author Response]

The following is the authors’ response to the original reviews.

**Public Reviews:**

**Reviewer #1 (Public review):**
Summary:The study by Lin et al. studies the role of EXOC6A in ciliogenesis and its relationship with the interactor myosin-Va using a range of approaches based on the RPE1 cell line model. They establish its spatio-temporal organization at centrioles, the forming ciliary vesicle and ciliary sheath using ExM, various super-resolution techniques, and EM, including correlative light and electron microscopy. They also perform live imaging analyses and functional studies using RNAi and knockout. They establish a role of EXOC6A together with myosin-Va in Golgi-derived, microtubule- and actin-based vesicle trafficking to and from the ciliary vesicle and sheath membranes. Defects in these functions impair robust ciliary shaft and axoneme formation due to defective transition zone assembly.Strengths:The study provides very high-quality data that support the conclusions. In particular, the imaging data is compelling. It also integrates all findings in a model that shows how EXOC6A participates in multiple stages of ciliogenesis and how it cooperates with other factors.Weaknesses:The precise role of EXOC6A remains somewhat unclear. While it is described as a component of the exocyst, the authors do not address its molecular functions and whether it indeed works as part of the exocyst complex during ciliogenesis.

We sincerely thank Reviewer 1 for the thoughtful evaluation of our manuscript and the constructive comments provided. We are especially grateful for the recognition of the quality and significance of our imaging data and the comprehensive model we propose regarding EXOC6A’s role in ciliogenesis. We did not address the function of other components of the exocyst complex during ciliogenesis. However, in our biochemical analyses, Myosin‑Va specifically co‑immunoprecipitated with EXOC6A but not with other exocyst subunits tested (EXOC5 and EXOC7) (Fig. 4E) indicating a selective interaction between EXOC6A and the Myo‑Va transport machinery.

**Reviewer #2 (Public review):**
Summary:The molecular mechanisms underlying ciliogenesis are not well understood. Previously, work from the same group (Wu et al., 2018) identified myosin-Va as an important protein in transporting preciliary vesicles to the mother vesicles, allowing for initiation of ciliogenesis. The exocyst complex has previously been implicated in ciliogenesis and protein trafficking to cilia. Here, Lin et al. investigate the role of exocyst complex protein EXOC6A in cilia formation. The authors find that EXOC6A localizes to preciliary vesicles, ciliary vesicles, and the ciliary sheath. EXOC6A colocalizes with Myo-Va in the ciliary vesicle and the ciliary sheath, and both proteins are removed from fully assembled cilia. EXOC6A is not required for Myo-Va localization, but Myo-VA and EHD1 are required for EXOC6A to localize in ciliary vesicles. The authors propose that EXOC6A vesicles continually remodel the cilium: FRAP analysis demonstrates that EXOC6A is a dynamic protein, and live imaging shows that EXOC6A fuses with and buds off from the ciliary membrane. Loss of EXOC6A reduces, but does not eliminate, the number of cilia formed in cells. Any cilia that are still present are structurally abnormal, with either bent morphologies or the absence of some transition zone proteins. Overall, the analyses and imaging are well done, and the conclusions are well supported by the data. The work will be of interest to cell biologists, especially those interested in centrosomes and cilia.Strengths:The TEM micrographs are of excellent quality. The quality of the imaging overall is very good, especially considering that these are dynamic processes occurring in a small region of the cell. The data analysis is well done and the quantifications are very helpful. The manuscript is well-written and the final figure is especially helpful in understanding the model.Weaknesses:Additional information about the functional and mechanistic roles of EXOC6A would improve the manuscript greatly.

We sincerely thank Reviewer 2 for the thoughtful and encouraging evaluation of our work. We are grateful for the recognition of the strengths of our study, including the quality of the TEM micrographs, the rigor of our imaging and data analysis, and the clarity of our manuscript and proposed model.

We have expanded our analyses in the revised manuscript to better define EXOC6A’s contribution to ciliary function. Specifically, we examined the trafficking of two critical ciliary membrane-associated proteins: GPR161, a G-protein-coupled receptor involved in Sonic hedgehog (Shh) signaling, and BBS9, a core component of the BBSome complex essential for ciliary membrane protein transport. Our new data (Fig. 7C) show that both GPR161 and BBS9 fail to localize to the cilium in EXOC6A knockout cells, in contrast to wild-type controls where their ciliary localization is robust. This new evidence significantly strengthens the understanding of EXOC6A’s role.

**Reviewer #3 (Public review):**
Summary:Lin et al report on the dynamic localization of EXOC6A and Myo-Va at pre-ciliary vesicles, ciliary vesicles, and ciliary sheath membrane during ciliogenesis using three-dimensional structured illumination microscopy and ultrastructure expansion microscopy. The authors further confirm the interaction of EXOC6A and Myo-Va by co-immunoprecipitation experiments and demonstrated the requirement of EHD1 for the EXOC6A-labeled ciliary vesicles formation. Additional experiments using gene-silencing by siRNA and pharmacological tools identified the involvement of dynein-, microtubule-, and actin in the transport mechanism of EXOC6A-labeled vesicles to the centriole, as they have previously reported for Myo-Va. Notably, loss of EXOC6A severely disrupts ciliogenesis, with the majority of cells becoming arrested at the ciliary vesicle (CV) stage, highlighting the involvement of EXOC6A at later stages of ciliogenesis. As the authors observe dynamic EXOC6A-positive vesicle release and fusion with the ciliary sheath, this suggests a role in membrane and potentially membrane protein delivery to the growing cilium past the ciliary vesicle stage. While CEP290 localization at the forming cilium appears normal, the recruitment of other transition zone components, exemplified by several MKS and NPHP module components, was also impaired in EXOC6A-deficient cells.Strengths:(1) By applying different microscopy approaches, the study provides deeper insight into the spatial and temporal localization of EXOC6A and Myo-Va during ciliogenesis.(2) The combination of complementary siRNA and pharmacological tools targeting different components strengthens the conclusions.(3) This study reveals a new function of EXOC6A in delivering membrane and membrane proteins during ciliogenesis, both to the ciliary vesicle as well as to the ciliary sheath.(4) The overall data quality is high. The investigation of EXOC6A at different time points during ciliogenesis is well schematized and explained.Weaknesses:(1) Since many conclusions are based on EXOC6A immunostaining, it would strengthen the study to validate antibody specificity by demonstrating the absence of staining in EXOC6A-deficient cells.(2) While the authors generated an EXOC6A-deficient cell line, off-target effects can be clone-specific. Validating key experiments in a second independent knockout clone or rescuing the phenotype of the existing clone by re-expressing EXOC6A would ensure that the observed phenotypes are due to EXOC6A loss rather than unintended off-target effects.(3) Some experimental details are lacking from the materials and methods section. No information on how the co-immunoprecipitation experiments have been performed can be found. The concentrations of pharmacological agents should be provided to allow proper interpretation of the results, as higher or lower doses can produce nonspecific effects. For example, the concentrations of ciliobrevin and nocodazole used to treat RPE1 cells are not specified and should be included. More precise settings for the FRAP experiments would help others reproduce the presented data. Some details for the siRNA-based knockdowns, such as incubation times, can only be found in the figure legends.Taken together, the authors achieved their goal of elucidating the role of EXOC6A in ciliogenesis, demonstrating its involvement in vesicle trafficking and membrane remodeling in both early and late stages of ciliogenesis. Their findings are supported by experimental evidence. This work is likely to have an impact on the field by expanding our understanding of the molecular machinery underlying cilia biogenesis, particularly the coordination between the exocyst complex and cytoskeletal transport systems. The methods and data presented offer valuable tools for dissecting vesicle dynamics and cilium formation, providing a foundation for future research into ciliary dysfunction and related diseases. By connecting vesicle trafficking to structural maturation of an organelle, the study adds important context to the broader description of cellular architecture and organelle biogenesis.

We sincerely thank Reviewer 3 for the thorough and thoughtful assessment of our manuscript. We greatly appreciate the recognition of the strengths of our study, including the use of advanced microscopy techniques, complementary functional tools, and the conceptual contributions regarding EXOC6A's role in vesicle trafficking and membrane remodeling during ciliogenesis.

Below, we detail how we have addressed the specific suggestions for improvement:

(1) Validation of EXOC6A Immunostaining Specificity

To directly address the reviewer’s concern regarding antibody specificity, we have included new control immunofluorescence panels in Figure S3E-F, which show a complete loss of EXOC6A signal in two independent knockout (KO) clones. These data confirm the specificity of the EXOC6A antibody used throughout the study and reinforce the accuracy of our localization analyses at different stages of ciliogenesis.

(2) Addressing Potential Clone-Specific or Off-Target Effects

To ensure that the observed phenotypes are attributable to EXOC6A loss and not due to off-target effects, we performed parallel analyses using two independent KO clones, all of which exhibited identical defects in ciliogenesis, including arrest at the ciliary vesicle stage and impaired cilia assembly (Fig. S3C-D).

In addition, we conducted rescue experiments by re-expressing EXOC6A in the KO background, which effectively restored ciliogenesis. Quantitative analysis of the rescue data has been added to the revised manuscript (Figure S6B), providing further support that the observed phenotype is specifically due to EXOC6A deficiency.

(3) Expanded Methodological Details

- A detailed protocol for co-immunoprecipitation experiments, including lysis conditions, antibody concentrations, and washing steps.

- The precise concentrations and treatment durations for all pharmacological agents used, including ciliobrevin and nocodazole.

- Comprehensive details on the siRNA-mediated knockdowns, including oligonucleotide sequences, transfection reagents, and incubation durations.

**Recommendations for the authors:**

**Reviewing Editor Comments:**
After further consultation, all 3 reviewers agreed that this is an important study with highquality data, in particular the imaging data. They also considered most of the evidence convincing, but overall they termed it "solid" for two main reasons: first, they would have liked to see a validation of the EXOC6A antibody specificity, and second, they suggest that you demonstrate for at least key experiments the phenotypes with a second KO clone, to exclude clonal effects. In principle, rescue would be suited to address this, but the issue here is that the presented rescue is not very robust.

We sincerely thank the Editor and all reviewers for their constructive and thoughtful evaluation of our manuscript. We are especially grateful for the recognition of the highquality imaging data, the experimental rigor, and the significance of our findings to the field of ciliogenesis.

We fully acknowledge the two principal concerns raised during further consultation: (1) the need for validation of EXOC6A antibody specificity, and (2) the importance of confirming the phenotypes in an independent knockout clone to exclude clonal artifacts. We have taken both of these points seriously and have now addressed them through additional experiments and analyses, as detailed below:

(1) Validation Using Independent Knockout Clones

To rigorously validate antibody specificity and eliminate the possibility of clonal variation, we have characterized a second independent EXOC6A knockout (KO) clone. We confirmed complete loss of EXOC6A expression in both clones using three orthogonal approaches: genotyping, immunoblotting, and immunofluorescence (Fig. S3). Both KO clones exhibit indistinguishable phenotypes, including arrest at the ciliary vesicle stage and impaired cilia formation (Fig. S3D).

(2) Rescue Phenotype Validation with Statistical Significance

In response to concerns about the robustness of the rescue, we have now included statistical analysis of the rescue experiments. A two-tailed Student’s t-test comparing ciliogenesis between the EXOC6A KO and rescue (GFP-EXOC6A re-expression) conditions shows a statistically significant improvement (p = 0.0041) (Fig. S6B). While we acknowledge that the rescue is partial—likely due to limitations of overexpression systems—the statistically significant recovery provides strong genetic evidence that the phenotypes are specific and reversible. These data are now included in the revised Figure S6.

(3) Functional Consequences of EXOC6A Loss on Ciliary Membrane Protein Trafficking

To further strengthen the mechanistic conclusions, we expanded our study to include the trafficking of two functional ciliary membrane proteins. We show that in EXOC6A KO cells, both BBS9 (a component of the BBSome complex) and GPR161 (a GPCR involved in Shh signaling) fail to enter the cilium. These results suggest that EXOC6A is required not only for early structural events in ciliogenesis, but also for establishing a competent transition zone, critical for ciliary membrane protein recruitment. These findings are detailed in the revised Figure 7C and corresponding Results.

We believe that these additional experiments and clarifications directly address the concerns and significantly strengthen the robustness and impact of our study.

The reviewers also made additional suggestions regarding functional and mechanistic insights that would strengthen the manuscript even further.
**Reviewer #1 (Recommendations for the authors):**
(1) The authors should include control IF panels for the specificity of the EXOC6A stainings at the various ciliogenesis stages using the KO cell line.

We thank the reviewer for this important suggestion. We have now included the requested immunofluorescence (IF) control panels to validate the specificity of the EXOC6A antibody. As shown in the newly added Figure S3, EXOC6A immunofluorescence signal is completely absent in EXOC6A knockout (KO) cells at CV (Fig. S3E) and cilia membrane (Fig. S3F) stages, whereas robust and stage-specific signals are observed in wild-type cells. These results confirm the specificity of the endogenous EXOC6A staining used throughout the study and validate the spatiotemporal localization patterns reported in the main figures.

(2) It would be informative to compare EXOC6A KO and RNAi to determine whether the only partially impaired ciliogenesis phenotype may be a consequence of cellular adaptation.

We appreciate the reviewer’s concern regarding potential cellular adaptation or clonespecific effects. To address this, we examined the ciliogenesis phenotype in two independent EXOC6A KO clones generated using distinct sgRNA targeting strategies. As shown in Figure S3, two independent KO clones displayed a highly consistent phenotype characterized by a pronounced arrest at the ciliary vesicle (CV) stage and a significant reduction in mature cilium formation.

The reproducibility of this phenotype across multiple independently derived clones strongly argues against clonal variability or long-term adaptive compensation as the underlying cause. Instead, these results support the conclusion that the observed ciliogenesis defects are a direct and specific consequence of EXOC6A loss.

(3) It remains unclear whether EXOC6A's function in ciliogenesis is part of the exocyst complex. This is currently implied by the context in which it is introduced and discussed, although the authors avoid any direct statement about this. Do the authors observe similar phenotypes by knocking down any other exocyst subunit? In any case, this issue should be discussed.

We thank the reviewer for raising this conceptual point. This study did not explore the functions of other components of the exocytosis complex during ciliogenesis, which warrants further investigation in the future. However, in our biochemical analyses, Myosin ‑Va specifically co‑immunoprecipitated with EXOC6A but not with other exocyst subunits tested (EXOC5 and EXOC7) (Fig. 4E) indicating a selective interaction between EXOC6A and the Myo‑Va transport machinery.

**Reviewer #2 (Recommendations for the authors):**
To clarify the roles of EXOC6A in ciliogenesis, I suggest the following:(1) Myo-Va is involved in both the intracellular and extracellular ciliogenesis pathways. The authors show that EXOC6A has a role in the intracellular ciliogenesis pathway. Does it also participate in the extracellular pathway?

We thank the reviewer for this insightful question. Given that Myo-Va functions in both intracellular and extracellular ciliogenesis pathways, it is indeed plausible that EXOC6A may also participate in the extracellular pathway. However, the current study was specifically focused on elucidating the molecular mechanisms of intracellular ciliogenesis using RPE1 cells, which exclusively undergo this pathway. Assessing EXOC6A’s role in the extracellular pathway would require the use of specialized models (e.g., polarized epithelial cells such as MDCK or IMCD3), which fall beyond the scope of this manuscript.

(2) In the live imaging movies (Fig 3C, 3D, supp movie 4 and 5), the authors observe tubular structures and puncta with EXOC6A and conclude that these are dynamic vesicles/membranes. While the movies are suggestive of membrane-like behavior, it would be helpful to show that these puncta and tubules have membrane, perhaps by astaining with a membrane dye.

We appreciate the reviewer’s suggestion to validate the membrane identity of EXOC6Apositive structures. While we did not perform membrane dye staining in the current study, we agree this approach would provide additional confirmation. Nevertheless, the dynamic behaviors observed in our live-cell imaging—including membrane-like tubulation, fusion, and fission—strongly support the interpretation that EXOC6A puncta and tubules

(3) It is unclear how the EXOC6A tubules and vesicles are delivered, and the extent to which MyoVa plays a role. The authors co-label EXOC6A and MyoVa in Supp Fig 2, but EXOC6A dynamics seem very different here, as compared to Fig 3D - there are fewer tubules and puncta and less movement of either tubules or puncta between time points. Does expression of MyoVa decrease EXOC6A membrane dynamics? Or is it required for EXOC6A membrane dynamics?

We thank the reviewer for this observation. The apparent differences in EXOC6A dynamics between Supplementary Figure 2 and Figure 3D most likely reflect cell-to-cell variability in dynamic behavior, which is common in live-cell imaging. Both figures were derived from the same stable cell line co-expressing EXOC6A and Myo-Va-GTD. Moreover, our analysis shows that Myo-Va-GTD overexpression does not suppress EXOC6A dynamics, nor is it required for membrane remodeling per se. However, Myo-Va is essential for EXOC6A recruitment to the ciliary vesicle, as shown by the loss of EXOC6A localization in Myo-Va KO cells (Fig. 4A).

(4) The authors show that loss of EXOC6A affects the localization of some transition zone proteins. Does this subsequently lead to defects in transition zone function?

We agree with the reviewer that structural defects in the transition zone (TZ) should be linked to its function. To address this, we examined the localization of two wellcharacterized ciliary membrane-associated proteins: BBS9 and GPR161. Both proteins failed to localize to the cilia in EXOC6A knockout cells, despite proper recruitment in wildtype controls (Fig. 7C). Although we did not examine the exact functions of GPR161 and BBS9, our results suggest that the loss of EXOC6A may impair TZ function, particularly its gating capacity for membrane protein trafficking.

(5) Additional information about how the MKS proteins are regulated by EXOC6A would be helpful to understand the mechanisms by which EXOC6A builds the transition zone. Does EXOC6A directly bind to MKS proteins, or are the MKS proteins delivered by EXOC6A-containing vesicles during ciliogenesis?

We appreciate the reviewers' questions regarding the mechanistic relationship between EXOC6A and MKS module proteins. In this study, we did not explore the mechanism by which EXOC6A constructs the transition zone. This is an interesting topic worthy of further investigation in the future.

**Reviewer #3 (Recommendations for the authors):**
Recommended modifications:(1) The co-immunoprecipitation experiments suggest an interaction between EXOC6A and Myo-Va; however, the presence of a faint band in the IgG control raises some uncertainty. To reinforce this conclusion, the authors could demonstrate that the interaction is absent in the EXOC6A knockout cell line.

We thank the reviewer for this careful observation. We acknowledge the presence of a faint Myo‑Va signal in the IgG control lane. Myosin‑Va is a highly abundant cytoskeletal motor protein and can occasionally exhibit low‑level nonspecific binding to agarose beads during immunoprecipitation assays. Importantly, the Myo‑Va signal co‑immunoprecipitated with endogenous EXOC6A is substantially stronger and specifically enriched compared with the IgG control, supporting a specific interaction.

(2) Figure S5: The partial rescue of the EXOC6A phenotype is not entirely convincing. A statistical test to assess the significance of the observed differences may help to strengthen the authors' conclusion.

We appreciate the reviewer’s suggestion to validate the rescue experiment. We have now performed a pairwise two‑tailed Student’s t‑test comparing ciliogenesis efficiency between EXOC6A knockout cells and rescue cells expressing GFP‑EXOC6A. As shown in the revised Figure S6 (original Figure S5), re‑expression of EXOC6A resulted in a statistically significant recovery of ciliogenesis (p = 0.0041). While the rescue is partial—likely due to inherent limitations of plasmid‑based expression systems, including variable transfection efficiency and imperfect restoration of endogenous protein levels—the statistically significant improvement confirms that the ciliogenesis defect is specifically caused by EXOC6A loss. Figure S6 and its legend have been updated accordingly.

(3) A detailed description of the EXOC6A knockout strategy should be included.

The Method section has been expanded to include a comprehensive description of the CRISPR/Cas9 ‑ mediated EXOC6A knockout strategy, including sgRNA sequences, genomic target sites, and validation approaches. Additionally, we now include Figure S3, demonstrating complete loss of EXOC6A protein expression in two independent knockout clones, confirming the efficiency and specificity of the gene‑editing strategy.

(4) The labeling in Figure 6 is confusing; assigning a separate letter to each panel would improve clarity.

Figure 6 has been reorganized for clarity: the original panels have been subdivided and relabeled as 6A/6A’ and 6B/6B’, respectively. The figure legend and all corresponding references in the main text have been updated accordingly.

(5) Lines 109-112: The cell line used is not well described. While experts might understand that Dox is used to induce expression of the transgenes, this should be better explained for non-expert readers.

We have revised the text to clearly explain that doxycycline (Dox) is used to induce transgene expression via a Tet‑On inducible system. This clarification has been added to the main text.

(6) Line 180: replace "labels" with "structures".

We have revised the text as suggested.

(7) Line 189: the EXOC6A recruitment to the membrane structures seems to be occurring on a short timescale that should be specified. In this context, "immediately" appears unscientific.

We have revised the sentence to specify that EXOC6A recruitment occurs within seconds, based on our live‑cell imaging data, providing a more accurate temporal description.

(8) Lines 280-282: We recommend rewording to soften this statement. Actin and microtubule inhibitors affect the entire cytoskeletal network; more specific experiments would be required to assess whether the transport of vesicles is defective.

We have reworded the statement to indicate that the accumulation of these vesicles at the mother centrioles is highly sensitive to disruption of dynein or microtubules, suggesting that efficient transport of these vesicles may depend on the integrity of the microtubule network. However, more experiments are required to confirm this conclusion.

(9) Lines: 428-433: Similarly, we recommend rewording this statement as it presents the authors' current model, which is in line with the presented data but would require more rigorous investigation.

We have revised this section to describe the mechanism as a working model supported by our data, while acknowledging that further investigation will be required to fully establish the proposed hierarchy and molecular details.

Questions and comments to consider:(1) 15-30% of cells can form cilia-like structures in the EXOC6A KO cells, although membrane transport should be reduced. It would be interesting to investigate whether these cilia are only formed intracellularly and fail to reach the cell surface.

We thank the reviewer for this insightful question. Using both immunofluorescence and electron microscopy, we observed that a subset of ciliary membranes in EXOC6A KO cells do appear to fuse with the plasma membrane. However, due to the low frequency and heterogeneous morphology of these structures, we were unable to reliably quantify this population.

(2) In the Western blot shown in Figure 4, EXOC6A appears at multiple molecular weights when detected with the anti-EXOC6A antibody. Providing a possible explanation for this shift would be helpful.

We clarify that the apparent molecular weight shift likely results from gel distortion during electrophoretic separation. Importantly, the specificity of the major EXOC6A band was rigorously validated by its complete absence in EXOC6A knockout lysates, confirming that the detected signal corresponds to EXOC6A.

(3) The Western blot in Figure 5B is not fully convincing; including additional independent blots would be nice.

We thank the reviewer for this suggestion. Figure 5B has been replaced with a blot from an independent experiment, improving clarity and reproducibility.

(4) According to the materials and methods section, siRNA-mediated knockdown of targets was performed using a single siRNA per gene, which could result in off-target effects. It would be advised to use several different siRNAs for a single target to exclude off-target effects, cite references or, in case this has been done.

We appreciate this concern. The siRNAs used in this study were previously validated in our earlier work (Wu et al., Nat Cell Biol 2018), where both specificity and efficiency were rigorously tested. We have now explicitly cited this reference in the Materials and Methods section to justify the selection of these reagents.

(5) The abbreviation CFLEM is uncommon for correlative (fluorescence) light and electron microscopy; the authors should consider using the standard abbreviation CLEM.

We have replaced “CFLEM” with the standard term CLEM (Correlative Light and Electron Microscopy) throughout the manuscript and figure legends.

(6) The term "M-centriole" is uncommon and should at least be introduced. The use of the term "mother centriole" is recommended.

We have replaced “M‑centriole” with the standard term “mother centriole” throughout the manuscript and figures.